# Deamidation disrupts native and transient contacts to weaken the interaction between UBC13 and RING-finger E3 ligases

**Priyesh Mohanty, Rashmi Agrata, Batul Ismail Habibullah, Arun G S, Ranabir Das\***

National Centre for Biological Sciences, Tata Institute of Fundamental Research, Bengaluru, India

**Abstract** The deamidase OspI from enteric bacteria *Shigella flexneri* deamidates a glutamine residue in the host ubiquitin-conjugating enzyme UBC13 and converts it to glutamate (Q100E). Consequently, its polyubiquitination activity in complex with the RING-finger ubiquitin ligase TRAF6 and the downstream NF-κB inflammatory response is silenced. The precise role of deamidation in silencing the UBC13/TRAF6 complex is unknown. We report that deamidation inhibits the interaction between UBC13 and TRAF6 RING-domain (TRAF6$^{RING}$) by perturbing both the native and transient interactions. Deamidation creates a new intramolecular salt-bridge in UBC13 that competes with a critical intermolecular salt-bridge at the native UBC13/TRAF6$^{RING}$ interface. Moreover, the salt-bridge competition prevents transient interactions necessary to form a typical UBC13/RING complex. Repulsion between E100 and the negatively charged surface of RING also prevents transient interactions in the UBC13/RING complex. Our findings highlight a mechanism wherein a post-translational modification perturbs the conformation and stability of transient complexes to inhibit protein-protein association.

**\*For correspondence:**
rana@ncbs.res.in

**Competing interests:** The authors declare that no competing interests exist.

## Introduction

Several bacterial pathogens secrete effector proteins that inhibit or co-opt the Ubiquitin (Ub) pathway to suppress the immune response of the host cell (*Ashida et al., 2014*). The human pathogenic bacteria *Shigella flexneri* inactivates the host inflammatory Nf-κB signaling, responsible for inducing inflammatory cytokine responses during pathogen invasion (*Sanada et al., 2012*). In the early events of interleukin-dependent activation of Nf-κB signaling, the Ub-conjugating enzyme UBC13, and the Ub-ligase TRAF6 function together to synthesize both unanchored polyubiquitin chains, and anchored polyubiquitin chains on TRAF6 and its substrate NEMO (*Chen, 2005*). These chains serve as a scaffold to bring together TAK1/2 and IKK kinases, eventually leading to phosphorylation and activation of the IKK kinases, IκB degradation and nuclear translocation of transcription factor Nf-κB. To inactivate Nf-κB signaling, *Shigella flexneri* secretes a Type III effector called OspI, which functions as a deamidase (*Sanada et al., 2012*). OspI specifically targets a glutamine residue in UBC13 and converts it to glutamate (Q100E). Ubiquitination reactions of TRAF6 either with UBC13 in the presence of OspI or with mutant Q100E-UBC13 (dUBC13) show a significant drop in polyubiquitination activity (*Sanada et al., 2012*). However, the mechanism underlying inhibition of polyubiquitination by deamidation of UBC13 remains unclear.

Ubiquitination is a eukaryotic post-translational modification (*Komander and Rape, 2012*), wherein the last glycine residue in the C-terminal tail of Ubiquitin (Ub) is activated and covalently attached to a substrate lysine residue. Ubiquitination involves three steps: an initial activation and thioester conjugation by the Ubiquitin-activating enzyme (E1), followed by the thioester conjugation

**eLife digest** *Shigella* is a highly infectious group of bacteria that attack the human digestive tract, causing severe and often deadly diarrhoea, especially in children. There is currently no vaccine to protect against the disease, and some strains are also now resistant to antibiotics. People get infected by eating or drinking contaminated foods and water. After passing through the stomach, *Shigella* invades and then multiplies in the lining of the intestine, eventually causing tissue damage and irritation.

During this process, *Shigella* 'hides' from its host's immune system by blocking how intestinal cells respond to infection. Normally, infected cells send out chemical signals that act like a call for help, attracting specialised immune cells to clear the infection. In intestinal cells, two proteins called UBC13 and TRAF6 work together to switch on this response. Specifically, TRAF6 needs to bind to UBC13 for the switch to turn on.

Like many proteins, UBC13 is formed of thousands of atoms; some of these are organized in 'functional groups', a collection of atoms joined in a specific manner and with special chemical properties. During *Shigella* infection, the bacteria produce an enzyme that changes a single functional group (an amino group) at a specific location within UBC13 for a different one (an hydroxyl group).

Previous research showed that this could stop the immune response in intestinal cells, but the mechanism remained unknown. Mohanty et al. therefore set out to determine exactly how a change of so few atoms could have such a dramatic effect.

Biochemical studies using purified proteins revealed that *Shigella*'s alteration to UBC13 did not change its overall structure. However, the altered protein could no longer bind to its partner TRAF6. Theoretical analysis and computer simulations revealed that the normal binding process relies on a positively charged amino acid (one of the protein's building blocks) in UBC13 and a negatively charged one in TRAF6 being attracted to each other. *Shigella*'s substitution, however, introduces a second negatively charged amino acid in UBC13. This 'steals' the positively charged amino acid that would normally interact with TRAF6: the electrical attraction between the two proteins is disrupted, and this stops them from binding.

The work by Mohanty et al. reveals the exact mechanism *Shigella* uses to dampen its host's immune response during infection. In the future, this knowledge could be used to develop more effective drugs that would help control outbreaks of diarrhoea.

to the Ubiquitin-conjugating (E2) enzymes, and a final step in which the Ub is covalently attached to the substrate amino group. The last step is typically catalyzed by a class of Ubiquitin ligases (E3), which contain either a RING (Really Interesting New Gene)-finger domain, U-box domain or HECT domain (*Metzger et al., 2012*). The RING-finger/U-box domain stabilizes a catalytic-closed conformation of the flexible E2 ~Ub species and drastically enhances the rate of Ub conjugation to substrates (*Dou et al., 2012*; *Plechanovová et al., 2012*; *Pruneda et al., 2012*). The E2 UBC13 functions with several RING-finger E3s like TRAF6 to synthesize K63-linked poly-Ub chains that function to activate DNA repair or immune response (*Fukushima et al., 2007*). Apart from the E3s, UBC13 also binds a co-factor MMS2, which does not activate UBC13 but maintains the linkage specificity of the poly-Ub chains synthesized by UBC13 (*Branigan et al., 2015*).

In this study, we have investigated the mechanisms underlying the inactivation of UBC13 upon deamidation using NMR spectroscopy, molecular dynamics (MD) simulations, and in-vitro ubiquitination assays. We report that deamidation weakens the non-covalent interaction of UBC13 with RING-finger domain of TRAF6 (TRAF6[RING]), without perturbing UBC13 structure or the intrinsic enzymatic activity of UBC13. However, the underlying cause of reduced interaction is nonintuitive since Q100 is in the vicinity of UBC13/TRAF6[RING] interface but does not form any contact with the TRAF6[RING]. Further studies showed that deamidation disrupts the interaction between UBC13 and TRAF6[RING] by three mechanisms: i) A new intramolecular R14/E100 salt-bridge appears in dUBC13, which competes with a critical intermolecular salt-bridge in the native complex, ii) the salt-bridge competition also perturbs the UBC13/TRAF6[RING] transient complexes to inhibit association, and iii) transient repulsion between the negatively charged E100 and the negatively charged interface of TRAF6[RING]

reduces association between UBC13 and TRAF6$^{RING}$. The effect of each mechanism on the binding was confirmed by binding studies using appropriate substitutions in either UBC13 or TRAF6$^{RING}$. The impact of deamidation on transient interactions was also observed using another RING domain from RNF38, indicating that the mechanism could be ubiquitous for UBC13/RING complexes. Our study highlights that deamidation of residues that do not directly participate in the E2/E3 interaction but are close to the interface can effectively modulate the interaction by perturbing the native and transient intermolecular contacts. The mechanism of regulating protein-protein transient interactions by post-translational modifications could be at play in other quintessential signaling pathways.

## Results

### Deamidation abolishes the interaction between UBC13 and TRAF6$^{RING}$

Two different constructs of TRAF6 was used in this study. The isolated RING domain TRAF6$^{RING}$ (aa: 50–124) was a shorter construct. The longer construct included the RING domain and three ZF domains (TRAF6$^{RZ3}$, aa: 50–211). TRAF6$^{RING}$ interacts with UBC13 and with the donor Ub in the UBC13 ∼Ub conjugate, while the interaction of ZF domains with donor Ub further stabilized the complex (*Middleton et al., 2017*). dUBC13 had reduced polyubiquitination activity than UBC13, either with longer TRAF6$^{RZ3}$ or with shorter TRAF6$^{RING}$ (*Figure 1A and B*). Deamidation could have some allosteric effect at the active site or Ub-binding site, which may deactivate UBC13. However, a comparison of the extent of polyubiquitination by the UBC13/MMS2 heterodimer in the absence of E3 was similar between UBC13 and dUBC13, indicating that E2 activity is not altered upon deamidation (*Figure 1C*).

Deamidation could misfold UBC13 or inhibit the interaction between UBC13 and TRAF6$^{RING}$. These possibilities were examined by studying the impact of deamidation on the structure of UBC13 and its interaction with TRAF6$^{RING}$. $^{15}$N-labeled UBC13, dUBC13, and unlabelled TRAF6$^{RING}$ were expressed in *E. coli* and purified. Unlabelled TRAF6$^{RING}$ was titrated into a sample of $^{15}$N-UBC13, and the binding was detected by $^{15}$N-edited Heteronuclear Single Quantum Coherence (HSQC) NMR experiments (*Figure 1—figure supplement 1*). Perturbations due to the altered chemical environment upon ligand binding induce changes in the chemical shift of the backbone amide resonances. The chemical shift perturbations (CSP) plotted in *Figure 1D* shows that the significant perturbations in UBC13 occur in the α1-helix, and the loop between α$_{3-10}$ and α2-helix, which is the canonical UBC13 interface in the complex (*Figure 1G and H*). The resonance shifts can be plotted against the ligand: protein concentration, and fit to yield the dissociation constant ($K_d$) of interaction. The peak shifts in UBC13 titration spectra were fitted to yield a $K_d$ of 0.39 (±0.04) mM (*Figure 1I* and *Figure 1—figure supplement 1B*).

Backbone amide chemical shifts can report if the protein's fold is affected by the substitution/modification of a residue. The backbone chemical shifts of UBC13 and dUBC13 resonances were compared. The amide CSPs between UBC13 and dUBC13 indicated that apart from the residues immediately next to Q100 in sequence, only R14 and L15 of the N-terminal α1-helix are affected upon Q100E substitution (*Figure 1—figure supplement 2A*). The absence of major CSPs in the rest of dUBC13 indicated that deamidation did not change the fold of UBC13. Moreover, analysis of the backbone and C$_β$ chemical shifts in dUBC13 confirmed that the secondary structure was unperturbed (*Figure 1—figure supplement 2B*). Unlabelled TRAF6$^{RING}$ was then titrated into a sample of $^{15}$N-dUBC13, and the binding was detected by $^{15}$N-edited HSQC experiments (*Figure 1—figure supplement 3*). Negligible peak shifts were detected in dUBC13 even when the TRAF6$^{RING}$ was titrated up to 6-fold higher than dUBC13, indicating that TRAF6$^{RING}$ does not bind dUBC13 (*Figure 1E–H*). A rough estimate of the $K_d$ was obtained by comparing the CSPs at the UBC13 interface at the same protein: ligand concentration between UBC13 and dUBC13, which suggested the $K_d$ of 6.34 (±2.6) mM. If the reduced affinity is precisely due to the negative charge at residue 100, then substitution of Q100 with a neutral amino acid should not affect the affinity of the complex. TRAF6$^{RING}$ was titrated to Q100A-UBC13, and the measured $K_d$ was 0.54 (±0.07) mM (*Figure 1I*), which is similar to UBC13/TRAF6$^{RING}$. However, when Q100 was substituted with another acidic residue aspartate (D), the affinity dropped significantly (Kd ∼4.6 (±1.8)) mM, *Figure 1I*), confirming that substitution at Q100 with a negatively charged residue inhibited the interaction between UBC13 and

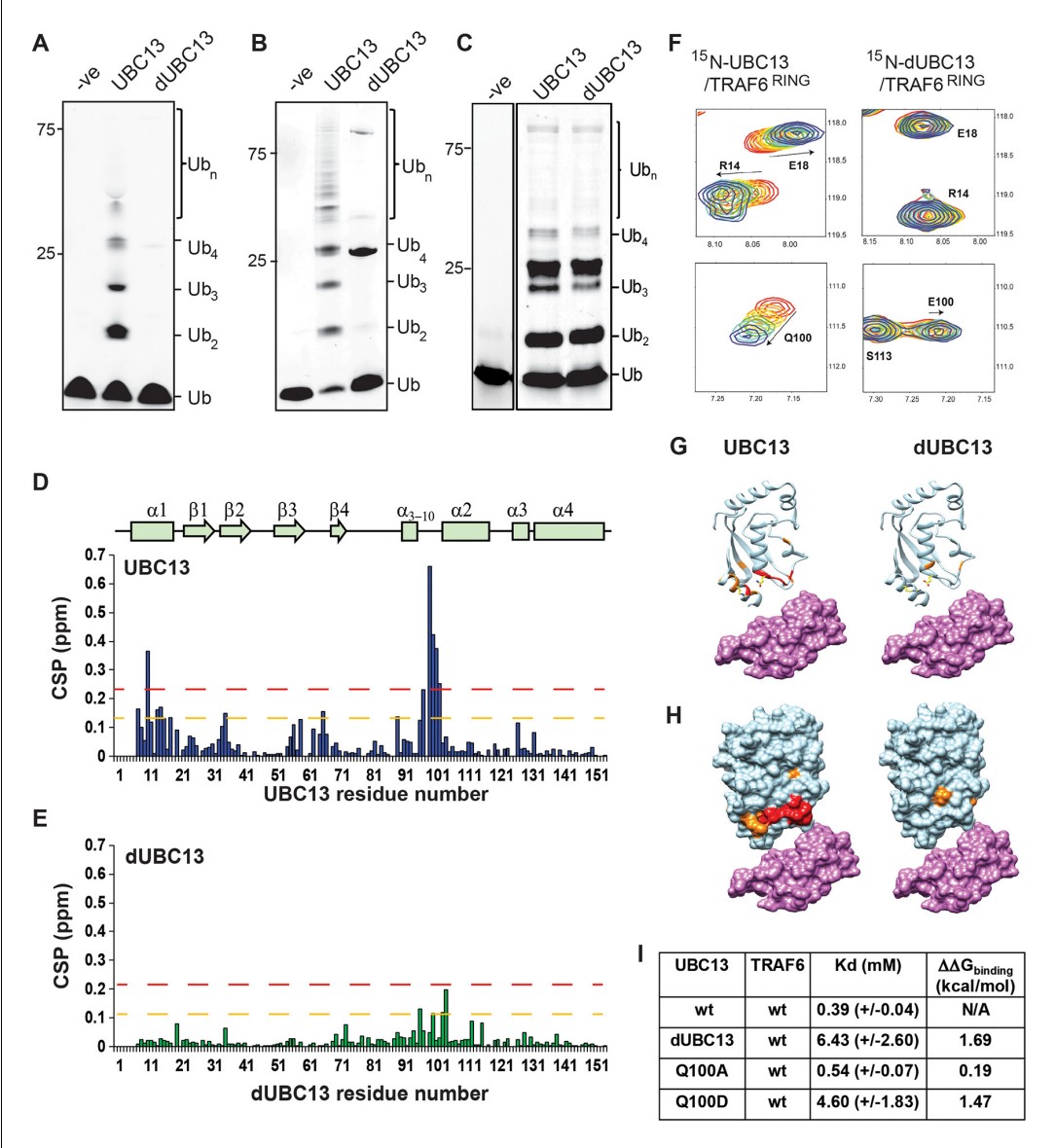

**Figure 1.** Interactions between UBC13 and TRAF6RING studied by NMR. (A) In-vitro ubiquitination reaction was carried out using TRAF6RZ3 as the E3 and UBC13 or dUBC13 as the E2 for 10 min. Mms2 was used as a co-factor in the reaction. The –ve lane is the same reaction without ATP. (B) In-vitro ubiquitination reaction was carried out on GST beads for 10 min using GST-TRAF6RING as the E3 and UBC13 or dUBC13 as the E2. Mms2 was used as a co-factor in the reaction. The –ve lane is the same reaction without ATP. (C) In-vitro ubiquitination reaction carried out for 30 min using UBC13 or dUBC13 as the E2 and Mms2 as its co-factor. (D)The CSPs for residues in UBC13 upon binding to TRAF6RING. The chemical shift perturbations (CSP) between the free and the bound form are calculated as CSP = $[(\delta^H_{free} - \delta^H_{bound})^2 + ((\delta^N_{free} - \delta^N_{bound}))^2]^{1/2}$, where $\delta^H$ and $\delta^N$ is the chemical shift of the amide hydrogen and nitrogen, respectively. The orange and red dashed lines correspond to Mean + SD and Mean + 2*SD, respectively. The secondary structure alignment of UBC13 against its sequence is provided above the plot. (E) The CSPs for residues in dUBC13 upon binding to TRAF6RING. The dashed lines are replicated from (D). (F) Two regions of the titration HSQC spectra are expanded to show UBC13, but not dUBC13 peaks shift upon titration with TRAF6RING. Significant CSPs were mapped on the UBC13 and dUBC13 structure both in the (G) ribbon and (H) surface representation. The UBC13 and dUBC13 are colored in light blue. The residues with CSPs above Mean + SD and Mean + 2*SD are colored in orange and red, respectively. The surface of TRAF6RING domain shown in magenta. The UBC13/TRAF6RING complex is modeled from PDB 3HCU. (I) The measured dissociation constants of UBC13 and its mutants with TRAF6RING are provided as Mean+/-SD. The difference of free energy of binding was calculated as $\Delta\Delta G_{binding} = RT\ln(K_d/K_d^{wt})$, where T is 298K, and wt is the wild type complex.

The online version of this article includes the following source data and figure supplement(s) for figure 1:

**Source data 1.** Source data of chemical shift perturbations against UBC13 residue numbers in the UBC13/TRAF6RING titration.

**Source data 2.** Source data of chemical shift perturbations against dUBC13 residue numbers in the dUBC13/TRAF6RING titration.

**Figure supplement 1.** Binding studies of UBC13/TRAF6RING interaction.

*Figure 1 continued on next page*

**Figure supplement 2.** NMR studies of UBC13 and dUBC13.
**Figure supplement 3.** Binding studies of dUBC13/TRAF6$^{RING}$ interaction.

TRAF6$^{RING}$. Altogether, deamidation inhibited the binding of TRAF6$^{RING}$ to UBC13, but neither changed the fold of UBC13 nor its intrinsic E2 activity.

## Deamidation triggers the formation of an intramolecular salt-bridge in UBC13

In the UBC13/TRAF6$^{RING}$ native complex structure, Q100 does not form any direct contact with TRAF6$^{RING}$ (*Figure 2A*). Hence, the mechanism underlying reduced binding due to deamidation of Q100 is nonintuitive. The surface electrostatic potentials calculated for UBC13 and TRAF6$^{RING}$ indicated charge complementarity at the interface (*Figure 3A*). The UBC13 interface was positively charged, while the TRAF6$^{RING}$ was negatively charged. Notably, Q100 lies in the vicinity of a network of electrostatic interactions involving helix-1 of UBC13 (R6 and R14) and the first zinc-coordination motif (D57 and E69) of the TRAF6$^{RING}$ (*Figure 2A*). Possibly, the introduction of an additional negative charge in UBC13 upon deamidation could perturb the interfacial electrostatic contacts to disrupt the interaction.

A conventional MD simulation (200 ns) of free dUBC13 was performed to investigate if deamidation alters any electrostatic interactions within UBC13. Interestingly, an intramolecular salt-bridge was observed between R14 and E100 in dUBC13 (*Figure 2B*), which correlates well with the backbone chemical shift perturbations observed at R14 upon deamidation (*Figure 1—figure supplement 2A*). The new salt-bridge involving R14 and E100 in dUBC13 was further investigated by NMR. $^{15}$N-edited HSQCs of the Arginine Nε-Hε groups were collected for both UBC13 and dUBC13, which showed shifted resonances for the arginine sidechains (R7, R14, and R102) around residue E100, indicating that these sidechain conformations are perturbed (*Figure 2C*). Unfortunately, the $^{1}$Hη atoms are invisible in the NMR spectra due to chemical exchange with the solvent at physiological pH, and millisecond timescale rotations around the Cζ-Nη/Cζ-Nε bonds (*Figure 2D*). Hence, detection of arginine sidechain-mediated interactions (hydrogen bonds and salt-bridges) through $^{1}$Hη atoms is difficult. However, such interactions can be inferred from the reduced mobility of their sidechains observed in $^{15}$Nη/ε(F1)-$^{13}$Cζ(F2) correlation spectra (*Yoshimura et al., 2017*). For example, R85 forms salt-bridge with D81 in UBC13, which reduces the sidechain mobility of R85 sidechain and gives rise to two separate resonances for R85 Nη atoms in the $^{15}$Nε/η-$^{13}$Cζ correlation spectra (*Figure 2E*). The side-chains of free arginines rotate faster and their corresponding Nη atoms have a single averaged resonance in the spectra (*Figure 2E*). The R14 Nη atoms had two separate resonances in dUBC13 but not UBC13 (*Figure 2F*, expanded in 2G), implying that R14 forms a new intramolecular salt-bridge in dUBC13, consistent with the MD simulations. The salt-bridge persisted in the Q100D-UBC13 (*Figure 2—figure supplement 1*), correlating well with its reduced affinity for TRAF6$^{RING}$.

## Deamidation triggers salt-bridge competition at the UBC13/TRAF6$^{RING}$ interface

At the interface of UBC13/TRAF6$^{RING}$, R14 of UBC13 forms an intermolecular salt-bridge with E69 of TRAF6$^{RING}$ (*Figure 2A*). The deamidation-induced intramolecular R14/E100 salt-bridge may interfere with the intermolecular R14/E69 salt-bridge to destabilize the complex. Conventional MD simulations were performed with both wild type (wt) and deamidated complex to test this hypothesis. The stability of the complex was determined by the rmsd of TRAF6$^{RING}$ with respect to the crystal structure, where higher rmsd implied instability. The wt complex was stable throughout the 200 ns trajectory with low rmsd (*Figure 2—figure supplement 2A*). The deamidated complex was unstable for a significant duration of 75 ns, wherein the rmsd of TRAF6$^{RING}$ increased substantially (*Figure 2—figure supplement 2B*).

Five interfacial contacts were chosen as reporters of the interaction between UBC13 and TRAF6$^{RING}$ (*Figure 2—figure supplement 2*). Among them, three were electrostatic (*Figure 2A* and *Figure 2—figure supplement 2A*), and the rest were hydrophobic (*Figure 2—figure supplement 2D*).

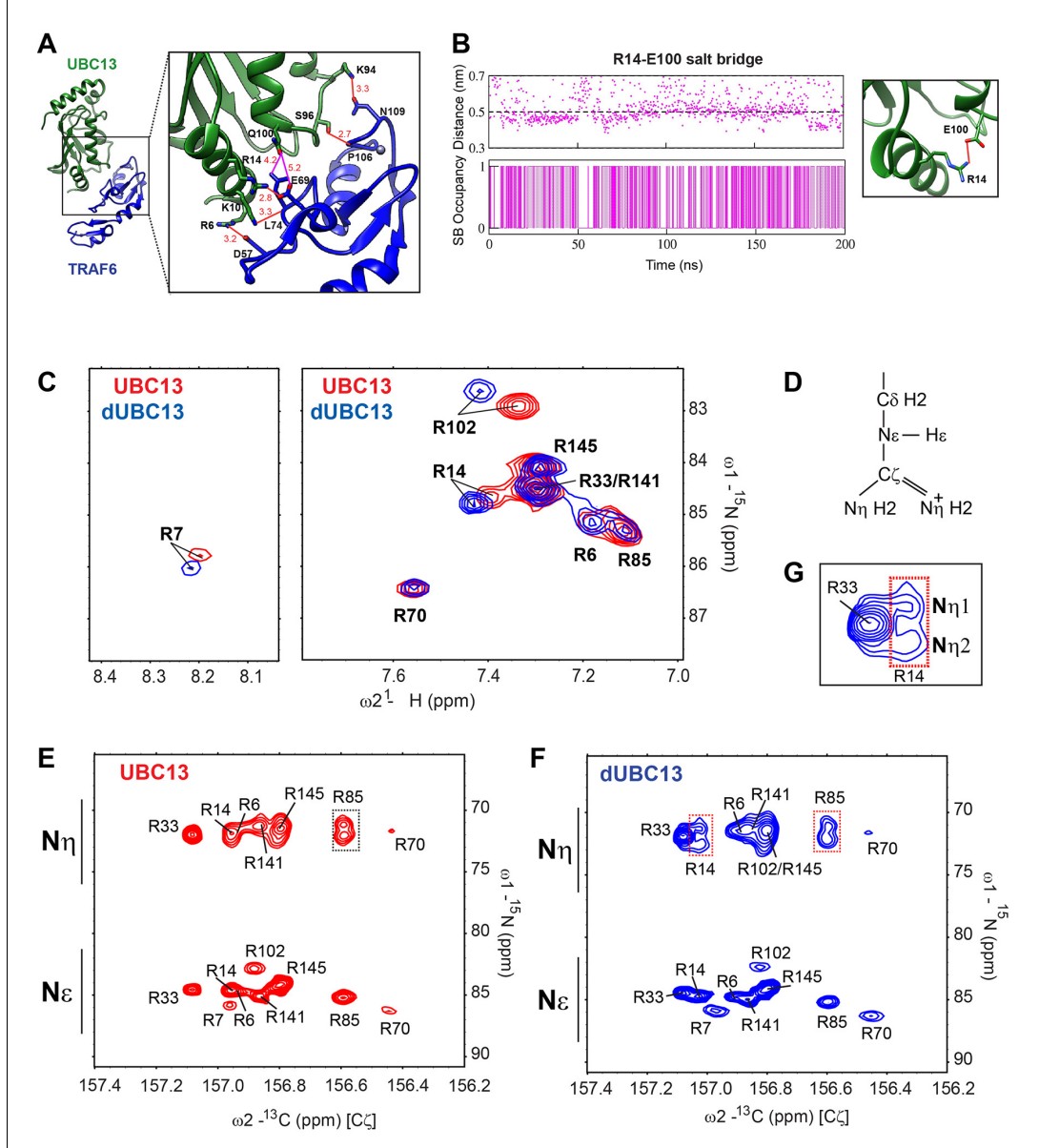

**Figure 2.** A new intramolecular salt-bridge forms upon deamidation of UBC13. (**A**) Crystal structure of the UBC13/TRAF6$^{RING}$ complex (PDB 3HCU, Left). Inset (right) indicates the position of Q100 and the network of salt-bridges/hydrogen bonds (red connecting lines) at the interface of UBC13/ TRAF6$^{RING}$ complex. The length of the bonds are provided in red (in Å). The distances from Q100 to its closest atom in TRAF6 residue L74, and to E69, are shown as magenta lines with distances in red. These distances are longer than the default criteria of contacts measured in UCSF Chimera. (**B**) The distance between R14-Cζ and E100-Cδ atoms against time in the conventional MD simulation of dUBC13 is given (top). The presence/absence of a salt-bridge based on a 0.5 nm cutoff value are digitized to a Markov chain, where the presence of a salt-bridge is one and absence is 0. The occupancy of the R14/E100 salt-bridge is 48%. (**C**) Overlay of UBC13 and dUBC13 $^{15}$N-$^{1}$H HSQC spectra zoomed around the Arginine Nε-Hε resonances shows that R7, R14, and R102 sidechain resonances shift upon deamidation. (**D**) Schematic of Arginine sidechain atoms. (**E**) and (**F**) are the $^{15}$Nε/η-$^{13}$Cζ correlation spectra for UBC13 and dUBC13, respectively. The $^{15}$Nε/η and $^{13}$Cζ resonance shifts are in the y- and x-axis, respectively. The R33 and R14 $^{15}$Nη resonances of dUBC13 are expanded in (**G**).

The online version of this article includes the following figure supplement(s) for figure 2:

**Figure supplement 1.** Intramolecular salt-bridge persisted in Q100D-UBC13.

**Figure supplement 2.** Conventional MD simulations of the UBC13/ TRAF6$^{RING}$ and dUBC13/ TRAF6$^{RING}$ native complexes.

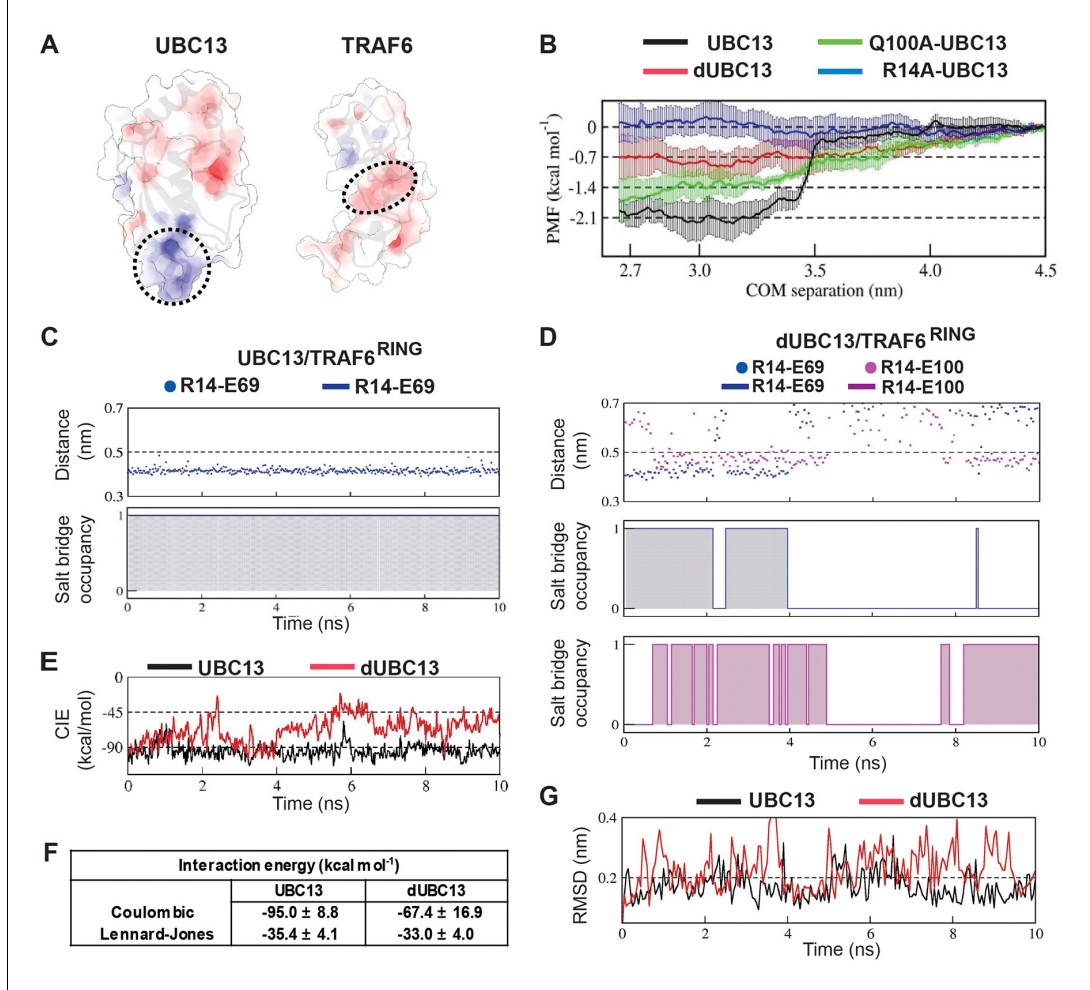

**Figure 3.** Deamidation induces salt-bridge competition to weaken UBC13/TRAF6^RING interaction. (**A**) Surface electrostatic potentials of UBC13 and TRAF6^RING. Circled regions indicate the complementary electrostatic surfaces on UBC13 and TRAF6^RING interfaces. The color scale ranges from −8 to +8 kT/e. (**B**) Potential of mean force (PMF) profiles as a function of COM separation along the x-axis for the association between UBC13 variants and TRAF6^RING. PMF profiles were calculated by averaging over five PMF profiles ranging from 2.5 ns to 10 ns. Error bars represent ± one standard error of the mean (SEM). (**C**) Stability and occupancy of the R14-E69 salt-bridge in the native window for the wild-type complex. D) Stability and occupancy of the R14-E69 and R14-E100 salt-bridge in the native window for the dUBC13/ TRAF6^RING complex. In (**C**) and (**D**), distance plots indicate the distance between R14 Cζ and E69/E100 Cδ atoms. The salt-bridge occupancy plots were generated as in *Figure 2B*. The salt-bridge occupancies are provided in *Supplementary file 1-table S5*. (**E**) Coulombic interaction energies (CIE) between UBC13 wild-type/Q100E and TRAF6^RING in native windows (mean COM separation = 2.7 nm) from the US simulations. The table in (**F**) reports the Mean ± SD of the interaction energies over 10 ns. (**G**) RMSD of TRAF6^RING (aa:70–109) against time in the UBC13 and dUBC13 complexes.

The online version of this article includes the following figure supplement(s) for figure 3:

**Figure supplement 1.** PMF profile for the Barnase-Barstar complex and CIE calculations for the UBC13 complexes at 3.5 nm US window.

The other interfacial contacts were mostly unstable in simulations and were omitted from our analysis (*Figure 2—figure supplement 2G and H*). No major instability was observed in the hydrophobic contacts between UBC13 and dUBC13 (*Figure 2—figure supplement 2E and F*). However, concurrent to the high rmsd in dUBC13 complex, the R14/E69 intermolecular salt-bridge disrupted and the R14/E100 intramolecular salt-bridge formed, suggesting that salt-bridge competition might destabilize the native dUBC13 complex (*Figure 2—figure supplement 2B*, *Video 1*). The occupancy of the R14/E69 intermolecular salt-bridge dropped from 99% in the wt complex to 40% in the dUBC13 complex (*Supplementary file 1-table S1*).

The Umbrella Sampling (US) method was used to estimate the stability of the deamidated complex (*Figure 3*). The US method quantifies the energy required ($\Delta G_{PMF}$) to form the native complex.

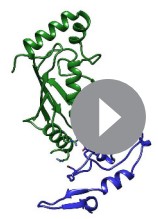

**Video 1.** Destabilization of the dUBC13/TRAF6$^{RING}$ native complex observed in a conventional MD simulation due to the formation of the R14-E100 intramolecular salt-bridge. This movie shows a change in the native intermolecular orientation indicated by an increase in RMSD of TRAF6$^{RING}$ beyond 0.5 nm, as shown in *Figure 2—figure supplement 2B*. Disruption of the R14-E69 salt-bridge occurs at ~ 25 ns followed by the formation of R14-E100 salt-bridge from ~ 75 to 150 ns. During this period, the S96-P106 contact is also lost. Residues R6/R14/S96/E100 in UBC13 (Green) and D57/E69/P106 in TRAF6$^{RING}$ (Blue) are shown in stick representation. A black, dashed-line is drawn between R14 (Cζ) and E100 (Cδ) which indicates the presence/absence of an intramolecular salt-bridge depending on the length of the line.

https://elifesciences.org/articles/49223#video1

An unstable complex is associated with higher energy. As a control, US simulations were performed on the Barnase/Barstar complex (*Table 1* and *Supplementary file 1-table S2*, *Figure 3—figure supplement 1A*). The PMF profile and $\Delta G_{PMF}$ of the complex are consistent with its high stability (*Hoefling and Gottschalk, 2010*; *Wang et al., 2010*). The US simulations were then carried out for wt, dUBC13, Q100A-UBC13, and R14A-UBC13 complexes (*Table 1*, *Supplementary file 1-table S2*, *Figure 3—figure supplement 1B*). The PMF profile of wt complex decreased sharply by 2.0 kcal/mol at 3.5 nm and subsequently plateaued (*Figure 3B*). The sharp drop in the PMF profile of the wt complex was due to attractive electrostatic interactions, which were absent in the other complexes (*Figure 3—figure supplement 1C and D*). In contrast, $\Delta G_{PMF}$ of dUBC13 complex gradually reduced by only 0.7 kcal/mol, suggesting that deamidation reduces the stability of the complex. The difference in $\Delta G_{PMF}$ between the deamidated and wt complex , $\Delta\Delta G_{PMF}$ = 1.37 kcal/mol was of the same order as the decrease in binding energy observed by NMR titrations, $\Delta\Delta G$ = 1.69 kcal/mol (*Figure 1I*). The $\Delta G_{PMF}$ of Q100A-UBC13/TRAF6$^{RING}$ is similar to the wt complex (*Table 1*).

The stability of the intermolecular contacts was compared between UBC13 and dUBC13 complexes at the native US window. While the R14/E69 salt-bridge was stable throughout in the wt complex, it disrupted in the dUBC13 complex (*Figure 3C and D*, *Supplementary file 1-table S5*). The Coulombic Interaction Energy (CIE) is the sum energy of all the electrostatic interactions. Lower CIE values reflect attractive electrostatic interactions. The mean CIE was significantly higher in dUBC13 complex, suggesting that deamidation primarily inhibited electrostatic interactions (*Figure 3F*). The disruption of R14/E69 salt-bridge temporally correlates well with higher CIE and higher rmsd of TRAF6$^{RING}$ (*Figure 3E and G*). Interestingly, when the US simulations were repeated with R14A-UBC13, the $\Delta G_{PMF}$ was insignificant, indicating that R14-mediated interactions are essential for the binding (*Figure 3B*). Overall, the conventional MD and US simulations indicate that salt-bridge competition may sequester R14 away from the interface and reduce the binding energy of the complex.

## Salt-bridge competition enhances the dissociation of dUBC13/TRAF6$^{RING}$

Steered MD (SMD) simulations were performed to capture the effects of deamidation on the dissociation of the UBC13 and dUBC13 complexes. If a complex is unstable, less force and work will be

**Table 1.** Association PMF determined by Umbrella sampling MD.

| Complex | <$\Delta G_{PMF}$> (kcal/mol) |
| --- | --- |
| Barstar/Barnase | −12.60 (±0.76) |
| UBC13/TRAF6 | −2.03 (±0.27) |
| dUBC13/TRAF6 | −0.67 (±0.50) |
| Q100A-UBC13/TRAF6 | −1.64 (±0.49) |
| R14A-UBC13/TRAF6 | +0.04 (±0.38) |

*Standard error of mean is indicated in brackets.

required to dissociate it. Compared to the wt complex, less force and work were required to dissociate the dUBC13 complex, indicating reduced stability due to deamidation (*Figure 4A–C*, *Table 2* and *Supplementary file 1-table S3*). As TRAF6$^{RING}$ starts to dissociate, the intramolecular R14/E100 salt-bridge competes with the intermolecular R14/E69 salt-bridge and reduces its strength (*Figure 4D*, *Figure 4—figure supplements 1–2* and *Supplementary file 1-table S4*). The R14A-UBC13 complex, where the intermolecular R14/E69 salt-bridge is absent, was also less stable compared to the wt complex (*Table 2*).

The order in which the contacts disrupted during dissociation was compared in a typical SMD trajectory for both UBC13 and dUBC13 complexes (*Figure 5A* and *Videos 2* and *3*). In the wt complex, whereas the other contacts disrupted early, the R14/E69 intermolecular salt-bridge persisted till the complex dissociated completely (*Figure 5A–5C* and *Figure 5—figure supplement 1A*). Since the intermolecular salt-bridge was stable for a long duration in both the conventional and SMD simulations, it could be critical for the wt complex. However, in the trajectory of the dUBC13 complex, the intermolecular salt-bridge disrupted prematurely, and the intramolecular salt-bridge formed simultaneously (*Figure 5D–5F* and *Figure 5—figure supplement 1B*). Collectively, SMD suggested that R14/E69 formed a critical interfacial salt-bridge, and as the molecules start to dissociate, the R14/E69 and R14/E100 salt-bridges compete. The competition may contribute to the reduced stability and enhanced dissociation of the dUBC13 complex.

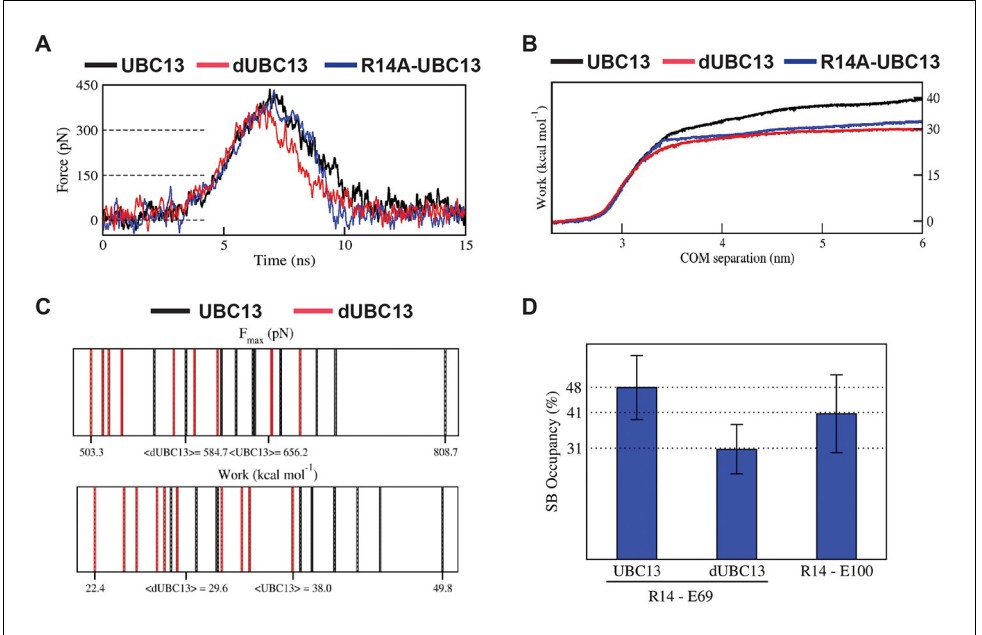

**Figure 4.** Steered MD of the UBC13/TRAF6$^{RING}$ and dUBC13/TRAF6$^{RING}$ complex. (**A**) Average force-extension profiles for wild-type and mutant complexes are plotted. The profiles were smoothened over 250 ps time intervals. (**B**) The plot of average work against COM separation indicated the cumulative work done to separate wild-type and mutant complexes. (**C**) Distribution of Fmax (top) and unbinding work (bottom) values obtained from ten individual SMD trajectories for wild-type (black) and mutant (red) complexes are shown. In each plot, the minimum, maximum, and average (< >) values are indicated on the x-axis. Both plots reveal a shift in the range of Fmax and unbinding work towards lower values for the dUBC13/TRAF6$^{RING}$ complex, which correlate with the reduced binding. (**D**) Mean ± one standard error (SEM) of R14-E69/E100 salt-bridge (SB) occupancies averaged from 5 to 15 ns of ten individual SMD trajectories are shown. SEM for all three occupancies is within 11%.
The online version of this article includes the following source data and figure supplement(s) for figure 4:

**Source data 1.** Source data of average salt-bridge occupancies in the SMD simulations.
**Figure supplement 1.** R14-E69 salt-bridge dynamics across ten trajectories (T1-T10) during dissociation of the UBC13/TRAF6$^{RING}$ complex by SMD.
**Figure supplement 2.** R14-E69 (blue)/R14-E100 (cyan) salt-bridge dynamics across ten trajectories (T1–T10) during dissociation of the dUBC13/TRAF6$^{RING}$ complex by SMD.

**Table 2.** Fmax and unbinding work determined by Steered MD.

| Complex | $\langle F_{Max} \rangle$ (pN) | $\langle$ Work $\rangle$ (kcal mol$^{-1}$) |
|---|---|---|
| UBC13/TRAF6 | 466.7 (±31.5) | 38.1 (±2.2) |
| dUBC13/TRAF6 | 425.9 (±42.2) | 29.6 (±1.6) |
| R14A-UBC13/TRAF6 | 450.4 (±45.9) | 31.9 (±2.1) |

*Standard error of mean is indicated in brackets.

## Repulsive interactions between E100 and TRAF6$^{RING}$ destabilize the transient complex

A native protein complex forms through an intermediate species commonly referred to as the transient/encounter complex ensemble (*Schreiber et al., 2009*). The transient complexes are formed by long-range electrostatic interactions and have greater oriental freedom compared to the native complex (*Tang et al., 2006*). The effect of negatively charged E100 on the transient complex was estimated based on transient complex theory (*Alsallaq and Zhou, 2008*). The method uses the native complex structure to compute its association rate constant ($k_{on}$) from the electrostatic free energy ($\Delta G_{el}$) of the transient complex ensemble. Position of individual atoms within each input structure remained fixed in transient-complex theory calculations, and hence, the competition between salt-bridges was absent. Therefore, any difference between wt and deamidated complex were solely due to transient repulsion between E100 and the acidic residues of TRAF6$^{RING}$. The association rate constant of the wt complex reduced at high salt compared to low salt, confirming that electrostatic interactions are important for the association of wt transient complex (*Table 3*). The association rate constant reduced by three-fold for dUBC13/TRAF6$^{RING}$ at low salt, indicating that deamidation reduces the association of the transient complex. The difference between wt and dUBC13 complex is nominal at high salt, where the repulsive effects of E100 may be screened.

Multiple US windows also indicated repulsive transient interactions between E100 and TRAF6$^{RING}$ (*Figure 6*, *Figure 6—figure supplement 1A*). The molecular details of the repulsive effect could be observed in a typical US window (*Figure 6*). In the wt complex, a transient attractive interaction was observed between R14 in UBC13 and D57 in TRAF6$^{RING}$ (*Figure 6A and C*, *Supplementary file 1-table S5*). The wt complex was stable with low rmsd and low CIE values (*Figure 6B and D*). However, the R14/D57 interaction was absent in the dUBC13 complex, and it dissociated prematurely (*Figure 6B–D*, *Supplementary file 1-table S5*). In the initial timepoints of this trajectory, R14 and E100 simultaneously interacted with D57 (*Figure 6C and E*). While the R14/D57 interaction was attractive, the E100/D57 interaction was repulsive. The repulsive interaction probably destabilized the attractive interaction, and it disrupted soon after. The complex dissociated subsequently (*Figure 6B*). The instability of deamidated complex was also evident in other US windows (*Figure 6—figure supplement 1B–E*). Overall, electrostatic repulsion between E100 and the negatively charged interface of TRAF6$^{RING}$ was observed in the dUBC13 transient complexes, which could destabilize it.

## Transient contacts of R14 are destabilized in the UBC13/TRAF6$^{RING}$ complex

When the transient complex theory calculations were repeated for the R14A-UBC13 complex, the $k_{on}$ dropped significantly by six-fold at low salt (*Table 3*). Since repulsion due to E100 is absent in R14A-UBC13, the drop in $k_{on}$ suggests that R14 also forms transient contacts with TRAF6, which are important for the binding. US simulations also suggested that R14-mediated non-native transient interactions promote the association of the wt complex (*Figure 6A*). Unbiased-association simulations were performed on the complexes to confirm this observation. From these simulations, the probability of association and stability of native-like transient complexes were assessed (*Table 4*). Similar to Transcomp calculations (*Table 3*), the native-like association of the wt complex was 3-fold higher at low salt than high salt, confirming that formation of the transient complex was dependent on the long-range electrostatic interactions. The association and stability dropped by 2.3 fold in the dUBC13 complex at low salt, suggesting that deamidation reduces the formation of the native-like transient complex.

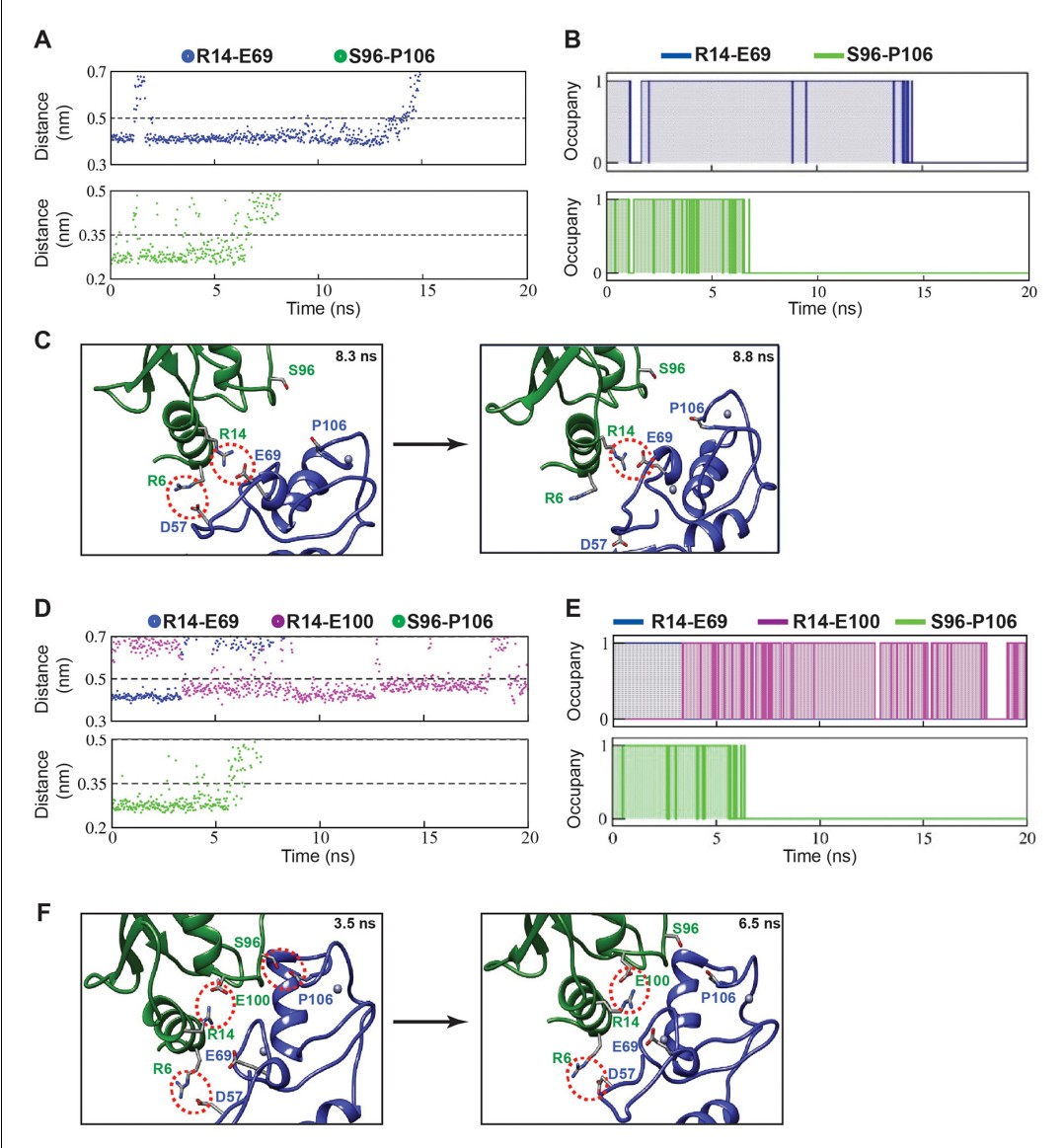

**Figure 5.** Pathway of UBC13/TRAF6$^{RING}$ complex dissociation from steered MD. (A) Top: The distance between R14-C$\zeta$ and E69-C$\delta$ atoms against time during SMD. Bottom: The distance between S96-O$\gamma$ and P106-O atoms against time during SMD. (B) Occupancy plots of (A) calculated as in *Figure 2B*. (C) Two snapshots from the trajectory in (A) are shown. S96-P106 contact is disrupted at 8.3 ns followed by the R6-D57 salt-bridge break at 8.8 ns. Red dotted circles indicate the polar contacts. (D) Same as (A) for the dUBC13/TRAF6$^{RING}$ complex. The distance between R14-C$\zeta$ and E100-C$\delta$ atoms against time is added here. (E) Occupancy plots of R14-E69 (blue), R14-E100 (magenta) and S96-P106 (green) contacts for the dUBC13/TRAF6$^{RING}$ complex. (F) Two snapshots from trajectory analyzed in (D) showed competition between R14-E100/E69 salt-bridges as the complex starts to dissociate at 6.5 ns.

The online version of this article includes the following figure supplement(s) for figure 5:

**Figure supplement 1.** Dynamics of R6-D57 salt-bridge and hydrophobic interactions in SMD.

The ensemble of transient complexes was compared between UBC13 and dUBC13 (*Figure 7*). The distribution of TRAF6$^{RING}$ positions around the helix 1 of UBC13 indicates that the TRAF6$^{RING}$ positions are tightly clustered around the UBC13 binding interface (*Figure 7A*). The projections of the transient complexes showed that a cluster indeed coincided with the final native complex. The positions of TRAF6$^{RING}$ are dispersed when it interacts with dUBC13, and none of the clusters coincide with the native complex (*Figure 7B*). Several TRAF6$^{RING}$ clusters around R14 were absent in the

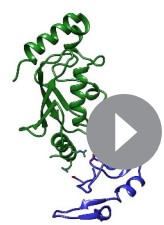

**Video 2.** Dissociation pathway of the UBC13/TRAF6$^{RING}$ native complex observed by steered MD. The movie shows the order of contact disruption during enforced dissociation of TRAF6$^{RING}$ from UBC13 (*Figure 5A–C*, *Figure 5—figure supplement 1A*). The loss of S96-P106 and hydrophobic interactions occur by ~9 ns. The R14-E69 salt-bridge persists until ~ 15 ns following which, complete dissociation occurs. Residues R6/R14/S96 in UBC13 (Green) and D57/E69/P106 in TRAF6$^{RING}$ (Blue) are shown in stick representation.
https://elifesciences.org/articles/49223#video2

**Video 3.** Dissociation pathway of the dUBC13/TRAF6$^{RING}$ native complex observed by steered MD. The movie shows the order of contact disruption during enforced dissociation of TRAF6$^{RING}$ from dUBC13 (*Figure 5D–F*, *Figure 5—figure supplement 1B*). The R14-E69 salt-bridge is lost within ~ 3 ns due to competition with E100. Loss of S96-P106 and hydrophobic interactions occur by ~11 ns. The R14-E100 intramolecular salt-bridge persists even after dissociation of the complex. Residues R6/R14/S96/E100 in UBC13 (Green) and D57/E69/P106 in TRAF6$^{RING}$ (Blue) are shown in stick representation. A black, dashed-line is drawn between R14 (Cζ) and E100 (Cδ) which indicates the presence/absence of an intramolecular salt-bridge depending on the length of the line.
https://elifesciences.org/articles/49223#video3

dUBC13 complex, indicating that the transient contacts between R14 and TRAF6$^{RING}$ are absent in dUBC13 (*Figure 7C and D*).

The energy landscape correlating the free-energy with rmsd of TRAF6$^{RING}$, and the minimum distance between interfaces shows multiple low-energy transient intermediates were present in the association pathway, which converged to the native complex (*Figure 7—figure supplement 1A*). In contrast, a few transient intermediates were present in the dUBC13 complex, and none existed in the native-like conformation (*Figure 7—figure supplement 1A*). The association trajectories were further analyzed to detect native-like complexes, which identified one trajectory at low salt for the wt complex (*Figure 7—figure supplement 2A*, *Figure 8A*). Here, TRAF6$^{RING}$ associated initially with UBC13 via the R6/D57 and R14/E69 salt-bridges (*Figure 8B*, *Video 4*). Subsequently, the other contacts formed, leading to a stable native complex (*Figure 8—figure supplement 1A*). No native complex was detected for the dUBC13 complex at low salt conditions (*Figure 7—figure supplement 2B*). At higher salt, where deamidation effects are potentially screened, a native-like complex was momentarily observed (*Figure 7—figure supplement 2B* and *Figure 8C*, *Video 5*). Here, a non-native transient R6/E69 salt-bridge and the S96/P106 hydrogen bond formed for a short period. However, the intramolecular R14/E100 salt-bridge was strong, whereas the intermolecular R14/E69 salt-bridge and other contacts were unstable (*Figure 8C and D*, and *Figure 8—figure supplement 1B*). To summarize, unbiased association simulations indicate that the critical transient contacts formed by R14 were absent in the deamidated complex, presumably due to the salt-bridge competition.

## Binding studies delineate the effects of deamidation on the UBC13/TRAF6$^{RING}$ complex

The MD simulations suggested that deamidation destabilized both the native and transient complexes. For simplicity, we assumed that the effect of salt-bridge competition on the transient and native complexes, and the repulsive effect of E100 on transient complexes are mutually exclusive. The difference in binding energy due to deamidation ($\Delta\Delta G^d$) is then

$$\Delta\Delta G^d = \Delta\Delta G_N^{SB} + \Delta\Delta G_T^{SB} + \Delta\Delta G_T^{rep}, \quad (1)$$

where $\Delta\Delta G_N^{SB}$ is the difference in binding energy due to salt-bridge competition in the native complex, $\Delta\Delta G_T^{SB}$ is due to salt-bridge competition in the transient complex, and $\Delta\Delta G_T^{rep}$ is the repulsive effect of E100 on the transient complex. NMR titrations were carried out using variants of UBC13 and TRAF6$^{RING}$ to validate the effect of each mechanism on the

**Table 3.** Electrostatic free energies and association rate constants ($k_{a0}$/$k_{on}$) of UBC13/TRAF6$^{RING}$ calculated using TransComp web server.

| Transient complex Ensemble | $k_{a0}$ ($10^6$ .M$^{-1}$s$^{-1}$) | 10 mM NaCl | | 100 mM NaCl | |
| --- | --- | --- | --- | --- | --- |
| | | $\Delta G_{el}$ (kcal mol$^{-1}$) | $k_{on}$ ($10^6$ .M$^{-1}$s$^{-1}$) | $\Delta G_{el}$ (kcal mol$^{-1}$) | $k_{on}$ ($10^6$ .M$^{-1}$s$^{-1}$) |
| UBC13/TRAF6 | 0.57 | −2.32 | 28.7 | −1.33 | 5.43 |
| dUBC13/TRAF6 | 0.63 | −1.59 | 9.36 | −0.99 | 3.43 |
| R14A-UBC13/TRAF6 | 0.62 | −1.16 | 4.42 | −0.62 | 1.78 |

binding energy of the complex. E69A-TRAF6$^{RING}$ removed the acceptor of the intermolecular salt-bridge and mimicked the effect of salt-bridge competition in the native complex ($\Delta\Delta G_N^{SB}$). When $^{15}$N-UBC13 was titrated with E69A-TRAF6$^{RING}$, the $K_d$ increased, and binding energy decreased (*Figure 9A*). The drop in binding energy compared to the wt complex gave the $\Delta\Delta G_N^{SB}$ = 0.82 kcal/mol. The R14A-UBC13 substitution mimicked the salt-bridge competition in both the native and the transient complexes, that is the drop in binding energy was $\Delta\Delta G_N^{SB}$ + $\Delta\Delta G_T^{SB}$. The titration of R14A-UBC13 with TRAF6$^{RING}$ determined a drop in the binding energy by 1.45 kcal/mol (*Figure 9A*). Given the value of $\Delta\Delta G_N^{SB}$ was determined above, $\Delta\Delta G_T^{SB}$ = 0.63 kcal/mol. Deamidation reduced the binding energy by 1.69 kcal/mol ($\Delta\Delta G^d$, *Figure 1I*). Since $\Delta\Delta G_N^{SB}$ + $\Delta\Delta G_T^{SB}$ = 1.45

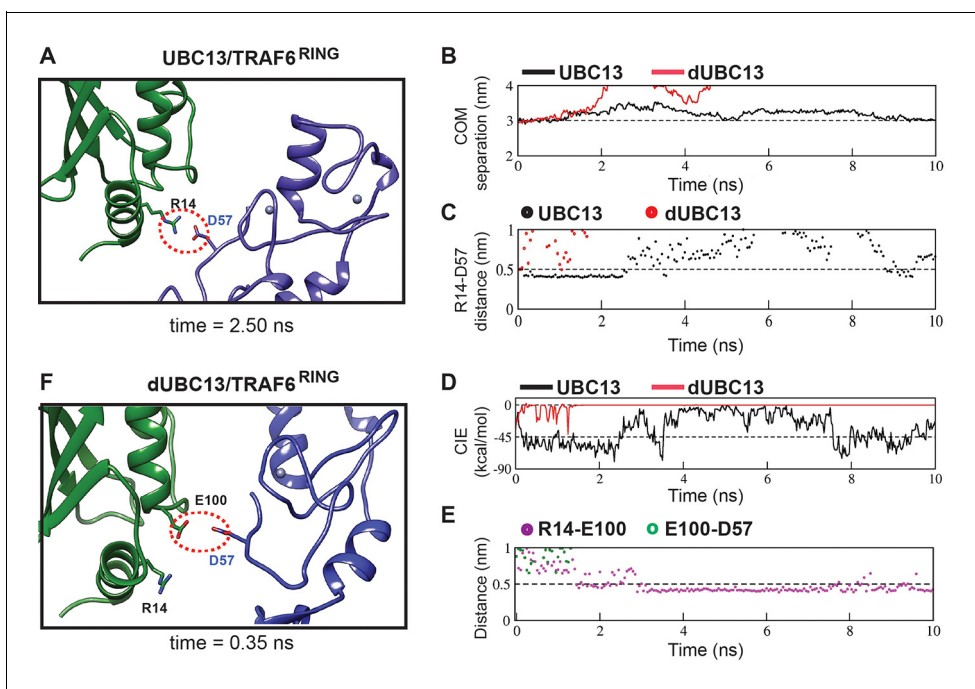

**Figure 6.** Analysis of transient complexes formed in the US simulations. The US window corresponding to 3 nm COM distance of separation was analyzed. (**A**) The transient complex of UBC13 and TRAF6$^{RING}$ at t = 2.50 ns is shown. A transient intermolecular salt-bridge between R14 and D57 is indicated by a red dotted circle. (**B**) The plot of COM separation against time for the UBC13/TRAF6$^{RING}$ and dUBC13/TRAF6$^{RING}$ complexes. (**C**) The distance between the R14-D57 transient contact is compared between the two complexes. The distances were measured between R14-Cζ and D57-Cγ atoms. (**D**) Coulombic interaction energy (CIE) plotted against time. (**E**) The contact distance of E100-D57 and R14-E100 contacts are shown against time for the dUBC13/TRAF6$^{RING}$ complex. The distances were measured between R14-Cζ, E100-Cδ, and D57-Cγ atoms. (**F**) The transient complex of dUBC13 and TRAF6 at t = 0.35 ns, where E100 contacted D57.

The online version of this article includes the following figure supplement(s) for figure 6:

**Figure supplement 1.** Analysis of interaction energy and COM separation between UBC13 and dUBC13 complexes from multiple umbrella sampling windows.

**Table 4.** Summary of unbiased-association MD simulations.

| | Association Trajectories* (%) | | | |
| --- | --- | --- | --- | --- |
| | 10 mM NaCl (low salt) | | 100 mM NaCl (high salt) | |
| | UBC13 | dUBC13 | UBC13 | dUBC13 |
| Native-like association | 53 (8) | 20 (3) | 27 (4) | 33 (5) |
| Non-native association | 47 (7) | 60 (9) | 40 (6) | 47 (7) |
| No association | 0 (0) | 20 (3) | 33 (5) | 20 (3) |
| | Mean percentage time (i.e., stability) of native-like association (%)** | | | |
| | 32 (9) | 14 (9) | 11 (7) | 11 (5) |

*The number of trajectories used is indicated in brackets.

**Standard error of mean is shown in brackets.

kcal/mol, $\Delta\Delta G_T^{rep}$ = 0.24 kcal/mol (*Equation 1*). The value of $\Delta\Delta G_T^{rep}$ agreed well with the repulsive effect calculated by transient complex theory calculations, $\Delta\Delta G_{el} = \Delta G_{el}^{UBC13} - \Delta G_{el}^{dUBC13}$ = 0.34 kcal mol$^{-1}$ (*Table 3*). Overall, titrations studies indicated that all three mechanisms have distinct contributions to the binding between UBC13 and TRAF6$^{RING}$.

## The effect of deamidation on transient interactions persist in the UBC13/RNF38$^{RING}$ complex

Analysis of the UBC13/RING structures shows that R14-mediated salt-bridges are common in these complexes (*Figure 9—figure supplement 1*). However, in a few complexes like UBC13/RNF4 and UBC13/RNF8, the R14 mediated salt-bridge was absent. A typical RING domain from the E3 RNF38 (RNF38$^{RING}$, aa: 387–465) was chosen to study the effect of UBC13 deamidation in such cases. RNF38$^{RING}$ can activate UBC13 to synthesize polyubiquitin chains (*Figure 9B*). Similar to TRAF6, the RNF38$^{RING}$ fails to activate dUBC13 (*Figure 9B*). Like RNF4 and RNF8, RNF38$^{RING}$ lacks a negatively charged residue at the region corresponding to E69 in TRAF6$^{RING}$ (*Figure 9—figure supplement 2A–C*). However, there are several acidic residues in the vicinity, similar to TRAF6$^{RING}$, which can form non-native transient interactions with UBC13 (*Figure 9—figure supplement 2B and C*).

NMR titrations measured the effect of deamidation on the interaction between UBC13 and RNF38$^{RING}$. When RNF38$^{RING}$ domain was titrated to $^{15}$N-UBC13 (*Figure 9C*), major CSPs were observed in the α1-helix, β3-β4 loop, and the loop between $\alpha_{3-10}$ and α1-helix (*Figure 9D and E*). The $K_d$ value of UBC13/RNF38$^{RING}$ interaction was 0.04 mM (*Figure 9I*). Considerable peaks shifts were also detected when RNF38$^{RING}$ was titrated to $^{15}$N-dUBC13 (*Figure 9F*). However, the CSPs in dUBC13 were reduced compared to UBC13 at the same protein: ligand stoichiometry, indicating reduced affinity (*Figure 9F–H*). Fitting of peak shifts against ligand: protein ratio yielded the $K_d$ to be 0.25 mM, which corresponds to a 6-fold drop in affinity. This could be due to the combination of the salt-bridge competition and repulsive interactions.

The backbone chemical shifts of RNF38$^{RING}$ resonances were assigned using $^{13}$C, $^{15}$N labeled RNF38$^{RING}$ sample, and standard triple resonance NMR experiments. The interface of RNF38$^{RING}$ was mapped by titrating unlabeled UBC13 to $^{15}$N-labeled RNF38$^{RING}$ (*Figure 9—figure supplement 2D*). Using the CSPs of UBC13 and RNF38$^{RING}$, an NMR-data driven structural model of UBC13/RNF38$^{RING}$ was determined by HADDOCK (*Figure 9—figure supplement 2E*). The structure showed that in the absence of a negative charged salt-bridge acceptor, R14 makes van der Waals contacts with M417 (*Figure 9—figure supplement 2F*). The R14A substitution reduces the binding energy of the native complex by 0.2 kcal/mol (*Vangone and Bonvin, 2015*), indicating it is not a hot-spot in this complex.

The titration was repeated with R14A-UBC13, which mimics the R14 sequestration by salt-bridge competition. The $K_d$ of R14A-UBC13/RNF38$^{RING}$ showed a 2.5-fold increase and reduced binding energy by 0.6 kcal/mol. Since the effect of R14A substitution on the native complex is 0.2 kcal/mol, the effect on the transient complex is 0.4 kcal/mol. Deamidation reduces the binding energy by 1.1 kcal/mol (*Figure 9I*). Given the salt-bridge competition reduces binding energy by 0.6 kcal/mol, the

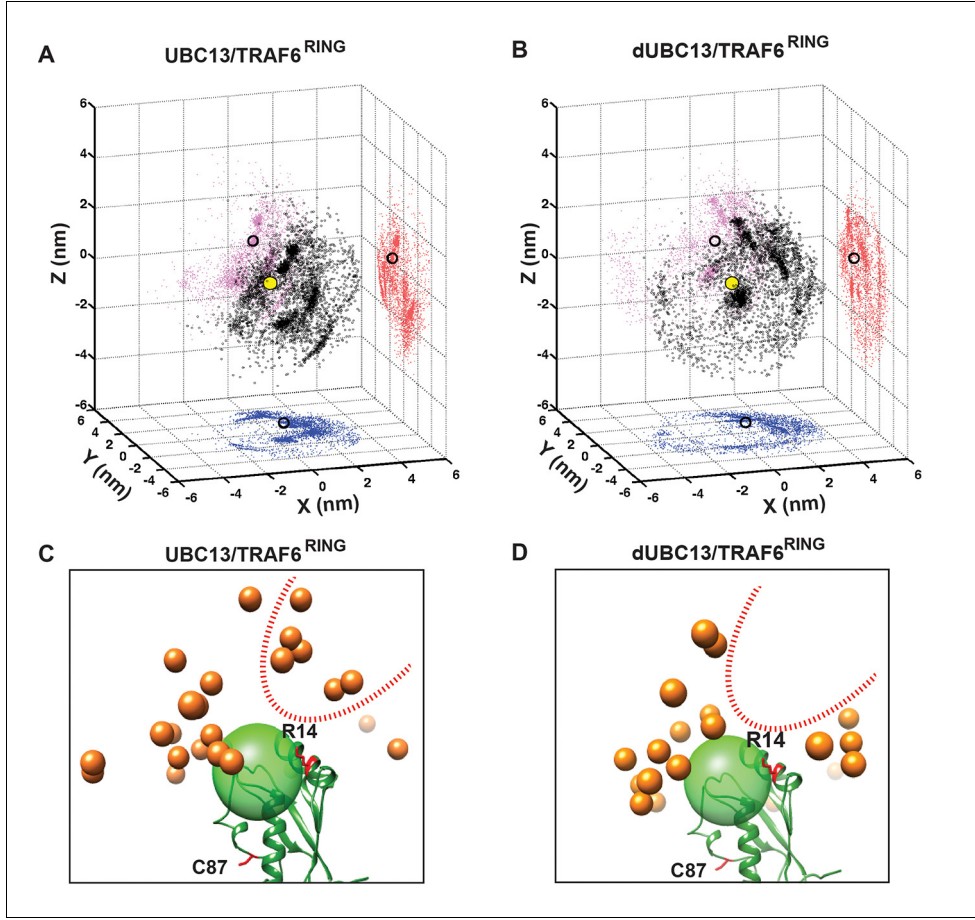

**Figure 7.** Transient complexes observed during unbiased association simulations. (**A**) Black open circles denote TRAF6$^{RING}$ (aa:70–109) centers of mass around UBC13 in the Cartesian coordinate space. Red, pink and blue colored symbols denote projections of the centers of masses on the coordinate planes. Projections of the centers of mass of TRAF6$^{RING}$ in the native complex are shown as black open circles on the coordinate planes. The center of mass of the UBC13 helix α1, which is at the center of UBC13 interface, is shown as a yellow sphere at the origin. (**B**) Same as in (**A**) for dUBC13. (**C**) TRAF6$^{RING}$ clusters calculated based on pairwise RMSD between UBC13/TRAF6$^{RING}$ complexes (cutoff = 0.45 nm) are shown as orange spheres. Only representatives for clusters with greater than ten structures are shown. The UBC13 is represented in ribbon and colored green. The center of mass of UBC13 interface is shown as a green transparent sphere with 1 nm radius. The sidechain of R14 is shown and colored red. (**D**) Same as (**C**) for dUBC13. The region near R14, where TRAF6$^{RING}$ clusters were absent in dUBC13 is shown as a red dotted curve in (**C**) and (**D**).

The online version of this article includes the following figure supplement(s) for figure 7:

**Figure supplement 1.** Free energy landscapes obtained from association MD (10 mM NaCl).

**Figure supplement 2.** RMSD analysis of UBC13/dUBC13 and TRAF6$^{RING}$ association trajectories.

repulsive effect of E100 on the transient complex is 0.5 kcal/mol. Overall, the effect of deamidation on perturbing the ensemble of transient complexes was verified in another RING domain, which indicated that the mechanism could be ubiquitous in UBC13/RING complexes.

## RING domains fail to bind and activate the dUBC13 ~Ub conjugate

If the RING domains bind weakly to dUBC13, their ability to activate the dUBC13 ~Ub conjugate should be compromised, which was tested by binding and activity assays. First, the conjugation of donor Ub to the UBC13 and dUBC13 were compared by an in-vitro conjugation reaction. Both the UBC13 and dUBC13 conjugated with the donor Ub with similar efficiency, indicating that deamidation does not hamper E2 ~Ub conjugation (*Figure 10A* and *Figure 10—figure supplement 1*). In the UBC13 ~Ub/TRAF6$^{RZ3}$ complex, the RING domain and ZF1 domain makes additional contacts

with donor Ub to stabilize the complex further. These contacts could rescue the loss in affinity between UBC13 and RING domain upon deamidation. Hence, the binding of UBC13 ~Ub and dUBC13 ~Ub to TRAF6$^{RZ3}$ was compared. While UBC13 ~Ub conjugate bound to TRAF6$^{RZ3}$, dUBC13 ~Ub conjugate failed to do so (*Figure 10B* and *Figure 10—figure supplement 1*). In a similar experiment, UBC13 ~Ub but not dUBC13 ~Ub bound to RNF38$^{RING}$ (*Figure 10C*). Altogether, the binding experiments confirmed that secondary interactions between Ub/RING or Ub/ZF1 could not compensate for the loss of primary interaction between UBC13 and RING.

The activation of UBC13 ~ Ub conjugates by the RING domains were measured by the kinetics of single-round Ub discharge from the UBC13 ~ Ub. UBC13 (or dUBC13) was first conjugated with K63A-Ub. Then the conjugation reaction was quenched, followed by the addition of Mms2, RING domains, and free Lysine. TRAF6$^{RING}$ could effectively discharge Ub from UBC13 ~ Ub, but not from dUBC13 ~ Ub (*Figure 10D,G,H*). Similarly, RNF38$^{RING}$ could discharge Ub from UBC13 ~ Ub but not from dUBC13 ~ Ub (*Figure 10E,I,J*). When the concentration of the RNF38$^{RING}$ was increased by five-fold to compensate for the reduced affinity of dUBC13/RING complex, the rate of Ub discharge improved, which confirmed that the cause of the inefficient discharge is the reduced affinity of dUBC13/RING interaction (*Figure 10K and F*).

The lysine of the acceptor Ub at the growing end of a poly-Ub chain (either unanchored or anchored on a substrate), attacks the E2 ~Ub active site to transfer the donor Ub. The kinetics of single-round di-Ubiquitin (Ub$_2$) synthesis was monitored to investigate if the deamidation of UBC13 affects the activity of UBC13 and its interaction with acceptor lysine at the active site. Here the lysine of the acceptor Ub mimics the acceptor lysine of growing poly-Ub chain. UBC13 (or dUBC13) was conjugated with donor K63A-Ub. Then Mms2, RNF38$^{RING}$ and acceptor D77-Ub was added in the reaction mix, which was then monitored over time to detect the synthesis of Ub$_2$. The rate of Ub$_2$ synthesis was rapid for UBC13 but slow for dUBC13 (*Figure 10L*). Kinetic analysis indicated that the reaction rate for the dUBC13/RING complex is similar to the rate of UBC13 in the absence of RING (*Figure 10M* and *Figure 10—figure supplement 2*), indicating that the catalytic effect of RING is absent in the dUBC13/RING complex. A comparison of the kinetic parameters between UBC13 and dUBC13 indicated that deamidation does not significantly affect the Km of the substrate, but drastically affects the Kcat of the reaction (*Figure 10N* and *Figure 10—figure supplement 2*). This confirms that the area around the active site is not perturbed by deamidation, as was observed by NMR (*Figure 1—figure supplement 2*). Furthermore, in the absence of a RING domain, the Kcat of UBC13 and dUBC13 is similar, confirming that the catalytic activity of UBC13 is unaffected upon deamidation, similar to the observation of *Figure 1C*. However, unlike UBC13, the Kcat of dUBC13 does not increase in the presence of RING, confirming that RING domains fail to bind and activate the dUBC13 ~Ub conjugate.

## Discussion

Protein deamidation can have significant functional consequences for innate immune signaling (*Zhao et al., 2016*), which justifies its emergence as a powerful tool employed by pathogenic bacteria to suppress the immune response (*Washington et al., 2013*). However, little is known about the molecular mechanism underlying deactivation of immune signaling by deamidation. The *Shigella flexneri* effector OspI suppresses the inflammatory response (NF-κB pathway) via deamidation of UBC13 at Q100. By an unknown mechanism, deamidation of Q100 to E100 inhibits the synthesis of K63-linked polyubiquitin chains by the UBC13/TRAF6 complex, preventing the downstream activation of NF-κB pathway. Q100 is present at the vicinity of UBC13/TRAF6 interface but does not form direct contact with TRAF6. We report that deamidation neither alters the fold nor the enzymatic activity of UBC13. NMR and kinetics studies indicate that deamidation does not alter the active site of UBC13. Instead, it reduces the interaction between UBC13 and RING domains to diminish the catalytic effect of the RING. The interaction with TRAF6 is inhibited because E100 in dUBC13 competes with an intermolecular R14/E69 salt-bridge at the dUBC13/TRAF6$^{RING}$ interface to form an intramolecular salt-bridge, which destabilizes the native complex. The new intramolecular salt-bridge also inhibits the long-range transient interactions formed by R14 with TRAF6$^{RING}$. In addition, repulsive interactions between E100 and the negatively charged interface of TRAF6$^{RING}$ destabilize the transient complexes. Cumulatively, these mechanisms reduced the binding energy between UBC13 and TRAF6$^{RING}$ by 1.69 kcal/mol. Consequently, its efficiency in activating the UBC13 ~Ub conjugate

drops by 6-fold (*Figure 10F*). The effect of deamidation is also manifested in another UBC13/RING complex. Deamidation reduced the rate of Ub-discharge and the rate of $Ub_2$ synthesis by several folds in the UBC13/RNF38$^{RING}$ complex (*Figure 10F and N*). These results provide first insights into the molecular mechanism behind the inactivation of the Ubiquitin pathway by bacterial deamidation.

For several E3s like RNF38, the RING domain can function as a monomer (*Buetow et al., 2015*). For others like TRAF6, the RING domains function as dimers (*Middleton et al., 2017*; *Yin et al., 2009*). Since deamidation perturbed the binding of UBC13 to both TRAF6 and RNF38, it is likely that deamidation inactivates UBC13 for both homo/hetero-dimeric RING-E3s. In dimeric RINGs like TRAF6, the proximal RING interacts with one unit of UBC13 ~ Ub, and the distal RING interacts with the donor Ub of another UBC13 ~ Ub. Additionally, for TRAF6 dimer, the first zinc-finger following the RING domain of one protomer also interacts with the donor Ub of the UBC13 ~ Ub bound to another protomer (*Middleton et al., 2017*; *Figure 10—figure supplement 3*). However, the binding between UBC13 and TRAF6$^{RING}$ is independent of dimerization (*Yin et al., 2009*), indicating that binding between UBC13 and proximal RING is primary. Indeed, deamidation inhibited the binding of UBC13 to both TRAF6$^{RING}$ and to the longer dimeric TRAF6$^{RZ3}$, confirming that secondary interactions between Ub/RING or Ub/zinc-finger cannot compensate for the loss of primary E2/RING interaction.

Though the interface of a protein complex has numerous interacting residues, a few hotspot residues contribute significantly to the binding energy than the rest (*Moreira et al., 2007*). Hotspot prediction algorithms reported R14 to be a hotspot at the UBC13/TRAF6$^{RING}$ interface (*Meireles et al., 2010*; *Zhu and Mitchell, 2011*). Analysis of all the UBC13/RING (or U-box) structures shows that intermolecular salt-bridges involving R14 are present in several of the UBC13/RING complexes (*Figure 9—figure supplement 1*). Binding studies by isothermal titration calorimetry measured that R14A substitution reduced the binding energy of UBC13/ZNRF1$^{RING}$ complex by ~1.2 kcal mol$^{-1}$ (*Behera et al., 2018*). Similarly, our NMR titrations measured that R14A substitution reduced the binding energy by ~1.4 kcal mol$^{-1}$ in the UBC13/TRAF6$^{RING}$ complex. Altogether, R14 appears to be a hotspot at the UBC13/TRAF6$^{RING}$ interface. Deamidation of UBC13 disrupts the salt-bridge between R14 and E69 due to competition from E100, which destabilizes the UBC13/TRAF6$^{RING}$ complex. Recently, the salt-bridge competition or 'theft' mechanism was observed in the binding switch of Raf Kinase Inhibitory Protein (RKIP) to either Raf-1 or GPCR-kinase 2 (*Skinner et al., 2017*). Phosphorylation of S153 in RKIP created a new salt-bridge between pS153 and K157, which disrupted the pre-existing salt-bridges involving K157 and resulted in the local unfolding of RKIP. The unfolding inhibited binding of RKIP to Raf-1 but promoted its binding to GPCR-kinase 2. Unlike RKIP, deamidation does not alter the local fold of UBC13. While intramolecular salt-bridges compete within RKIP, the competition is between an intermolecular and intramolecular salt-bridge in UBC13.

Besides the hotspots, electrostatic substitutions at the vicinity of the interface can considerably increase protein-protein association and affinity (*Selzer et al., 2000*). Such substitutions would shift the equilibrium between native and transient complexes (*Volkov et al., 2010*). The proper conformation of transient complexes is vital for two proteins to form a stable, productive complex (*Pan et al., 2019*; *Schilder and Ubbink, 2013*). How PTMs affect native protein-protein interactions observed in the ground state structure of the complex is well appreciated. However, little is known about how PTMs can affect higher energy transient complexes. PTMs like phosphorylation, eliminylation, and deamidation can change the surface electrostatics of a protein (*Ribet and Cossart, 2010*) to modulate the conformation and population of transient complexes significantly. By a combination of all-atom simulations and experimental data, this study provides molecular details of how PTMs can modulate transient protein-protein association to inhibit interaction. In the dUBC13/RNF38 complex, deamidation majorly alters transient association. Nonetheless, this causes a 10-fold drop in the catalytic activity of the complex (*Figure 10N*), indicating that modifying the transient protein-protein association can severely affect function.

The fate of the polyubiquitinated substrate depends on the linkage specificity of the conjugated polyubiquitin chain, which in turn depends on the specificity of the E2/E3 interaction. Rationally designed E2/E3 interactions can change the fate and function of cellular proteins (*van Wijk et al., 2009*). Hence, the interfacial contacts that determine E2/E3 specificities are a subject of intense research (*Christensen et al., 2007*; *Soss et al., 2011*; *van Wijk et al., 2009*; *van Wijk et al., 2012*). The strength and functionality of E2/E3 interactions have been typically tested by mutating the interfacial residues (*Christensen et al., 2007*; *Das et al., 2009*; *Das et al., 2013*). This work provides

evidence that transient interactions have an equally significant role in E2/E3 interaction and function. Rational design strategies will improve significantly by incorporating transient E2/E3 interactions.

In several cases, the native interactions cannot explain the functional implications of PTMs. This study introduces a new approach to understand the functional role of PTMs by considering their transient interactions. For example, spontaneous deamidation (non-enzymatic) is rampant in age-onset diseases, including neurodegenerative and ocular diseases. Deamidation of amylin accelerates its aggregation and amyloid formation (*Dunkelberger et al., 2012*). Crystallins deamidate with age, which promotes its aggregation to trigger cataract (*Pande et al., 2015*). Crystallin surfaces are highly charged, and possibly deamidation alters their charge distribution to promote non-native association leading to aggregation. However, the sites of deamidation on crystallins do not provide a clear mechanism of the process. Deamidation-induced modulation of transient protein-protein association may explain the tendency of these proteins to aggregate.

The bacterial effector OspI from *S. flexneri* presents a fascinating case, wherein a pathogen employs subtle mechanisms like salt-bridge competition and modification of transient protein-protein association, to produce a remarkable cumulative effect of inhibiting protein-protein interaction, polyubiquitination, and the host immune response. Such a mechanism could be a ubiquitous mode of regulating cellular pathways by PTMs.

# Materials and methods

## Key resources table

| Reagent type (species) or resource | Designation | Source or reference | Identifiers | Additional information |
|---|---|---|---|---|
| TRAF6 (*Homo sapiens*) | TRAF6$^{RING}$ (50-124) | Thermo Fischer Scientific, USA | | |
| UBC13 (*Homo sapiens*) | UBC13 | *Pruneda et al., 2012* | | |
| RNF38 (*Homo sapiens*) | RNF38$^{RING}$ (387-465) | *Buetow et al., 2015* | | |
| Strain, strain background (*Escherichia coli*) | BL21(DE3) | Invitrogen | | Protein Expression |
| Strain, strain background (*Escherichia coli*) | DH5α | Invitrogen | | DNA purification |
| Sequence based reagent | Oligo-dNTP | Sigma | | For site-directed mutagenesis of proteins |
| Commercial assay or kit | NEB HF cloning kit | New England Biolabs, Inc USA | | BamHI NotI Phusion HF (Pol) |
| Commercial assay or kit | DNA purification kit | Promega | Cat no: A4160 | DNA purification |
| Chemical compound, drug | N15- Ammonium Chloride, C13- D-Glucose Deuterium Oxide | Cambridge Isotope Laboratory, Inc | Cat no: NLM-467 CLM-1396 DLM-4 | Isotope enrichment media |
| Chemical compound, drug | NaCl, Tris-HCl, Glycerol, Triton, ATP, MgCl2, BME, EDTA | Sigma Aldrich | | Protein purification |
| Chemical compound, drug | Alexa Fluor Malemide 488 | Invitrogen | Cat no: A10254 | Protein Labelling |

*Continued on next page*

*Continued*

| Reagent type (species) or resource | Designation | Source or reference | Identifiers | Additional information |
|---|---|---|---|---|
| Software, algorithm | NMR Pipe MATLAB ImageJ GROMACS | *Delaglio et al., 1995*, https://www.mathworks.com/products/matlab.html *Schneider et al., 2012*, Nature Materials and methods 9 (7): 671–675. *Abraham et al., 2015* | | Data processing |
| Other | SYPRO Ruby stain | Invitrogen | Cat no: S21900 | Protein visualization |
| Other | Amylose bead | NEB | | |
| Other | Glutathione agarose | Thermo Fischer | | |

## Initial structures and molecular modeling

The interfacial contacts of UBC13/TRAF6$^{RING}$ were obtained from PDB entry 3HCU using the typical cut-offs in the contact analysis tool in UCSF Chimera (*Pettersen et al., 2004*). Initial structures of UBC13 and TRAF6$^{RING}$ for MD simulations and association rate constant ($k_{on}$) estimation were obtained from the PDB entry 3HCU (Chain A/B), which represents a model of the native complex. For MD simulations of TRAF6$^{RING}$, a C-terminal truncated model comprising of residues 50–148 was used. Substitutions were introduced by replacing existing sidechain with the best aligning rotamer from the Dunbrack rotamer library (*Dunbrack, 2002*) in UCSF Chimera (*Pettersen et al., 2004*). Electrostatic surface potentials of UBC13 and TRAF6$^{RING}$ were calculated using the Adaptive Poisson-Boltzmann Solver (*Baker et al., 2001*).

## General molecular dynamics (MD) simulation protocol

All simulation methodologies employed in the study were performed using the AMBER99SB-ILDN force field (*Best and Hummer, 2009*; *Hornak et al., 2006*; *Lindorff-Larsen et al., 2010*). Unbiased MD simulations were performed in GROMACS version 4.6.4, while biased simulations were performed using the pull code in Gromacs 5.1.2 (*Abraham et al., 2015*; *Hess et al., 2008*). All the acidic and basic residues apart from Histidines were modeled in their charged states. The Zinc AMBER force field (*Peters et al., 2010*) parameters were used to model the two zinc coordination sites within the TRAF6$^{RING}$ domain. The initial structures were solvated in an appropriate box using the TIP3P (*Jorgensen et al., 1983*) water model. The non-bonded ion parameters proposed by *Joung and Cheatham (2008)* were used to model Na$^+$ and Cl$^-$ ions in TIP3P water. Additional non-bonded parameter corrections proposed by Yoo and Aksementiev for cation-chloride (*Yoo and Aksimentiev, 2012*), amine-carboxylate (*Yoo and Aksimentiev, 2016b*) and aliphatic carbon-carbon (*Yoo and Aksimentiev, 2016a*) interactions were used for all simulations to eliminate overestimation of the strength of these interactions. A suitable number of counterions were added to neutralize the residual charge of the system, and additional ions were added to the box depending on the desired concentration. The electrically neutral, solvated system was then subjected to energy minimization using the steepest descent method for a maximum of 5000 steps until the maximum force on any atom was less than 1000 kJ mol$^{-1}$ nm$^{-1}$. Production simulations were performed under periodic boundary conditions at a temperature of 300 K, and 1 bar pressure (NPT ensemble) following equilibration carried out for 600 ps with a two fs time step. Temperature control was achieved using the v-rescaling thermostat (*Bussi et al., 2007*) for both equilibration ($T_c$ = 0.1 ps) and production ($T_c$ = 2.5 ps) steps. The Berendsen (*Berendsen et al., 1984*) ($T_p$ = 1 ps) and Parrinello-Rahman (*Parrinello and Rahman, 1981*) barostat ($T_p$ = 5 ps) were employed for pressure control during equilibration and production steps respectively. All bond lengths were constrained using the LINCS (*Hess, 2008*) algorithm. Virtual interaction sites were employed for hydrogen atoms (*Bjelkmar et al., 2010*), which permitted the use of a five fs time step. Short-range electrostatics and van der Waals interactions were calculated using a 1.0 nm cut-off. Long-range electrostatics were calculated using

the smooth Particle Mesh Ewald (PME) method (*Darden et al., 1993*; *Essmann et al., 1995*). An analytical dispersion correction was applied to approximate the effect of long-range van der Waals interactions.

## Analysis of conventional MD simulations

Simulations of free dUBC13 and UBC13/dUBC13 native complexes with TRAF6$^{RING}$ were performed using the general MD protocol described above. Salt-bridge occupancies were computed using a cut-off distance of 0.5 nm between arginine C$\zeta$ and glutamate C$\delta$ atoms using a python script. For unbiased-association simulations, UBC13/dUBC13 and TRAF6$^{RING}$ were separated along with the x component of the vector connecting their center of masses by 4 nm in a cubic box with an edge length of 12.6 nm. This results in a box volume equal to ~2000 nm$^3$ and the resulting concentration of each protein species equal to ~1 mM. Fifteen independent association runs (100 ns each with different initial atomic velocities) were performed at 10 mM NaCl and 100 mM NaCl concentrations.

Trajectories were analyzed every 250 ps using analysis scripts available within the Gromacs package. Minimum distances were measured between UBC13 and TRAF6$^{RING}$ to identify native-like, non-native, and non-associating trajectories using g_mindist. Native-like transient complex formation was considered to occur in a trajectory when the minimum distance between the sidechain nitrogen atoms of R6/K10/R14 (UBC13) and sidechain oxygen atoms of D57/E69 (TRAF6$^{RING}$) fell below 0.35 nm and persisted for more than eight ns. Among the remaining trajectories, minimum distances were measured between all UBC13 and TRAF6$^{RING}$ heavy atoms. Non-native and non-associating trajectories were then identified using the same distance and persistence period cut-off.

The spatial distribution of TRAF6$^{RING}$ COMs around UBC13/dUBC13 was obtained by calculating the displacement between COMs of C$\alpha$ atoms of helix-I in UBC13 and TRAF6$^{RING}$ (70-109) using g_dist and plotted for all fifteen association trajectories. RMSD-based structural clustering (*Daura et al., 1999*) with a cut-off of 0.45 nm was performed on pairwise rmsd matrices (determined using g_rmsd) of the combined trajectories (1.5 μs) using UBC13/dUBC13 (C$\alpha$ atoms) as a reference for superposition to obtain the UBC13/TRAF6$^{RING}$ transient complex clusters. The rmsd was computed for the backbone atoms of both UBC13 and TRAF6$^{RING}$ between all trajectory snapshots. The output structures of the complex were then superposed on UBC13 (PDB: 3HCU) to obtain the location of TRAF6$^{RING}$ cluster representatives. Native complex formation was identified by measuring the C$\alpha$ rmsd of TRAF6$^{RING}$ with respect to its position in the crystal complex for trajectories which exhibited the formation of native-like transient complexes using UBC13 (C$\alpha$ atoms) as a reference for superposition.

Two-dimensional free energy landscapes for UBC13/dUBC13 association with TRAF6$^{RING}$ were computed as a function of minimum distance (for salt-bridges between UBC13-R6/K10/R14 and TRAF6-D57/E69) and TRAF6$^{RING}$ rmsd using a python script. Bin width of 0.02 nm was used for both coordinates, and the free energy (Δ*G*) corresponding to each bin were determined using the relation:

$$\Delta G(R1, R2) = -K_B T \left[ \ln P_i - \ln P_{max} \right] \quad (2)$$

where R1/R2 are the two reaction coordinates, $k_B$ is the Boltzmann constant, T is the temperature, $P_i$ is the joint probability of R1/R2 in a given bin and $P_{max}$ is the max value of the probability. The lowest free energy state corresponds to ΔG = 0. Visualization of MD trajectories and analysis of the native contacts was performed using the MD analysis tool in UCSF Chimera.

## Umbrella sampling (US)

Umbrella sampling is an enhanced-sampling method (*Torrie and Valleau, 1977*) to determine free energy differences for state transformations by calculating the potential of mean force (PMF) as the function of a predefined reaction coordinate ($\xi$). This amounts to performing multiple, equilibrium MD simulations along with a range of $\xi$ values and determining the corresponding free energy ($A(\xi)$). At points along $\xi$ which are referred to as windows, a harmonic restraining potential ($w_i$) is added to the original Hamiltonian in order to restrain the system close to a target value ($\xi_i$):

$$E^b(r) = E^u(r) + w_i(\xi) \quad (3)$$

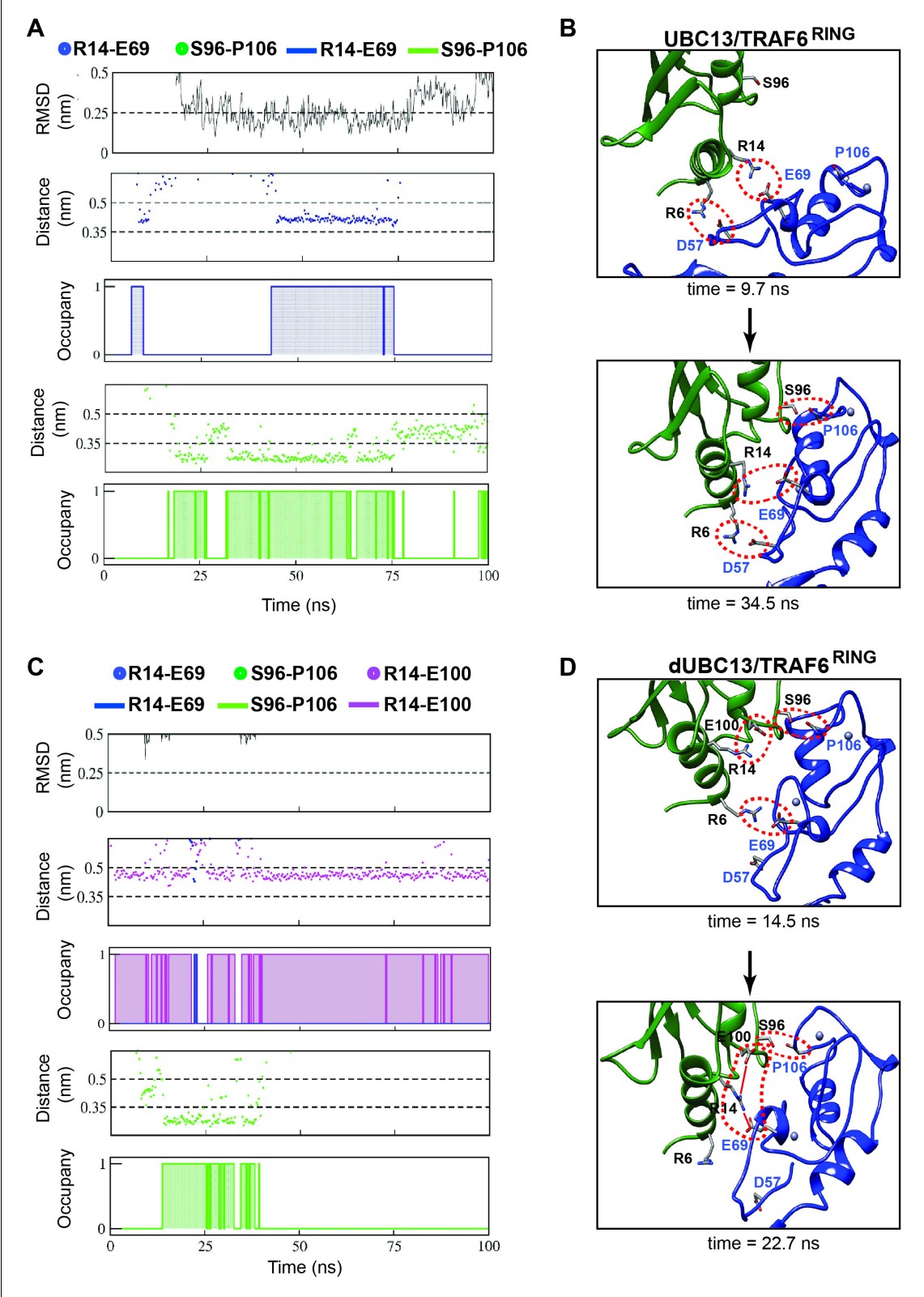

**Figure 8.** Association pathway of the UBC13/TRAF6 complex from unbiased MD trajectories. (A) and (C) shows the RMSD of TRAF6$^{RING}$, the distance between polar contacts and occupancies of polar contacts in UBC13/TRAF6$^{RING}$ and dUBC13/TRAF6$^{RING}$ trajectory, respectively. The occupancies are calculated as in *Figure 2B* and provided in *Supplementary file 1-table S6*. (B) Two snapshots from a trajectory in (A) is shown, where the initial contacts formed at 9.7 ns and the native complex formed at 34.5 ns. The polar contacts at the interface are shown by red dotted circles. (D) Two

*Figure 8 continued on next page*

*Figure 8 continued*

snapshots from a trajectory in (**B**) is shown, where the transient contacts formed at 14.5 ns, and the salt-bridge competition was observed at 22.7 ns (shown by red lines).

The online version of this article includes the following figure supplement(s) for figure 8:

**Figure supplement 1.** Dynamics of R6-D57 salt-bridge and hydrophobic interactions in association MD.

$$w_i(\xi) = k/2^*(\xi - \xi_i)^2 \tag{4}$$

In *Equation (2)*, $E(r)$ denotes the potential energy function (force field) of the system as a function of atomic coordinates ($r$). Subscripts b and u indicate biased and unbiased energies. $k$ *is* the force constant for the harmonic bias potential whose magnitude determines the range of $\xi$ values sampled within each window. The application of the bias potential particularly improves sampling in high energy regions along $\xi$ which are poorly sampled in conventional MD simulations. The modified energy function yields biased probability distributions ($P_i^b(\xi)$) of $\xi$ from which the unbiased, free energy ($A_i(\xi)$) of a window can be computed using the relation:

$$A_i(\xi) = -k_B T ln P_i^b(\xi) - w_i(\xi) + F_i \tag{5}$$

The value of $F_i$ needs to be determined for all windows to combine adjacent windows and obtain the global free energy profile ($A(\xi)$), which describes state transformation. The weighted histogram analysis method (WHAM) (*Kumar et al., 1992*) is widely used to determine optimal values of $F_i$ in an iterative fashion so as to obtain convergence of the free energy profile.

Similar to previous studies which utilized the US approach to study the energetics of protein-protein association, the center of mass (COM) separation along the x-axis was chosen as the reaction coordinate to calculate the PMF for Barnase/Barstar and UBC13/TRAF6 $^{RING}$ association. Barstar and UBC13 were translated along the x-axis of the vector joining the COM between their binding partners to positions separated by 0.1 nm intervals to achieve final separations of 4.0 and 4.5 nm respectively. Each window was simulated in a rectangular box of dimensions 12.6 nm x 9.0 nm x 9.0 mm following equilibration. During production MD (10 ns for each window), a harmonic restraining potential ($k$ = 1500 kJ mol$^{-1}$ nm$^{-2}$) was applied to the COM vector in each window and data were collected every 250 fs. The biased COM probability distributions from each window were analyzed based on WHAM and combined to obtain the global free energy (PMF) profile using the g_wham analysis tool (*Hub et al., 2010*). A total of 200 bins were used to construct a mean PMF profile from five independent time blocks (1.5 ns each from 2.5 to 10 ns) for each window (*Supplementary file 1-table S2*). A tolerance of $10^{-9}$ was used to check for convergence. The free energy of association ($\Delta G_{PMF}$) along the COM coordinate was determined by setting the free energy profile to zero in the unbound state (4.0 nm for Barnase/Barstar and 4.5 nm for UBC13/TRAF6$^{RING}$) and determining the value of the profile at mean COM separation in the native complex (2.3 nm for Barnase/Barstar and 2.7 nm for UBC13/TRAF6$^{RING}$) window during production MD. Coulombic interaction energies from the US windows were calculated using g_energy tool in Gromacs.

## Steered MD

Steered MD is a non-equilibrium, enhanced-sampling technique wherein a time-dependent biasing potential is applied on a selected atom/COM of atomic groups to induce perturbation in the system (*Grubmüller et al., 1996*). This is achieved by means of a moving dummy atom attached to the pull group via a stiff harmonic spring. This method has been previously employed to study the unfolding of protein domains and the dissociation of protein-ligand complexes. In this study, we carried out force-induced unbinding of TRAF6$^{RING}$ from UBC13 using the constant velocity pulling method along the x-axis to a COM separation of ~ 6 nm in 30 ns long simulations. The external force ($F_{ext}$) acting on the pull group as the simulation progresses with time ($t$) is defined by the relation:

$$F_{ext} = k(vt - x) \tag{6}$$

where $k$ is the spring constant, v is the pull velocity, and x is the displacement of the ligand from its initial position. The pull force was applied to COM of Cα atoms of the RING domain (residues 70–

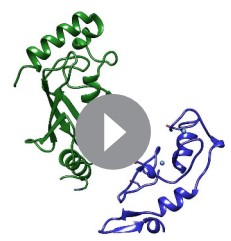

**Video 4.** Association pathway of UBC13/TRAF6$^{RING}$ (10 mM NaCl) observed by conventional MD. The movie shows the order of contact formation during the association of TRAF6$^{RING}$ and UBC13 (*Figure 8A–B*, *Figure 8—figure supplement 1A*). Initial association occurs through the formation of R6-D57 and R14-E69 salt-bridge. Subsequent formation of hydrophobic and S96-P106 contacts lead to the acquisition of the native orientation (top panel, RMSD ~ 0.3 nm). Residues R6/R14/S96 in UBC13 (Green) and D57/E69/P106 in TRAF6$^{RING}$ (Blue) are shown in stick representation.
https://elifesciences.org/articles/49223#video4

109) at a low pull rate of 0.125 nm ns$^{-1}$ (k = 1500 kJ mol$^{-1}$ nm$^{-2}$). The same box dimensions were used as in the case of umbrella sampling. The COM of Cα atoms of UBC13 was kept fixed by periodically removing its rotational and translation motion to promote dissociation of the complex. The cumulative work (W) done to separate complex was calculated using the relation:

$$W = \int F \cdot dx = 1/2 \sum (f_i + f_i)(X_{i+1} - X_i) \quad (7)$$

The work done over two successive simulation steps (i and i+1) is calculated as a product of the average force (f) multiplied by the displacement of the ligand between the steps. W is thus calculated by numerically integrating the work done between all successive steps of the simulation. The force and COM position of TRAF6$^{RING}$ was recorded every five ps. F$_{max}$ (rupture force) is defined as the maximum value of the force recorded in the Force-COM extension profile of an individual or averaged SMD trajectory. W was defined at a COM separation of 4.5 nm in the Work-COM extension profile as all native contacts were found to be disrupted at this COM separation. Both F$_{max}$ and W correlate with the binding affinity of the complex and hence allow for an analysis of the effect of mutations on the stability of the complex.

## Transient complex theory calculations

The association rate constants (k$_{on}$) for UBC13/dUBC13/R14A-UBC13 with TRAF6$^{RING}$ were calculated based on transient complex theory (*Alsallaq and Zhou, 2008*). The crystallographic complexes were provided as inputs to the Trans-Comp webserver (*Qin et al., 2011*). The server computes the association rate constant (k$_{on}$) from a basal association rate constant (k$_{a0}$) in the absence of electrostatic interactions and the electrostatic free energy (ΔG$_{el}$) of a pre-generated transient complex ensemble using the relation:

$$k_{on} = k_{ao} exp(-\Delta G_{el}/kT) \quad (8)$$

where k and T are the Boltzmann constant and temperature, respectively. The k$_{on}$ values for all complexes were computed at ionic strengths of 10 and 100 mM NaCl.

To generate the transient complex ensemble from the crystallographic complex, proteins are treated as rigid bodies and configurations are randomly sampled along six relevant degrees of freedom. Three degrees of freedom are for relative rotation and three for relative translation. Sampling is performed such that the distance between the center of the binding sites of the interacting proteins remains within a suitable

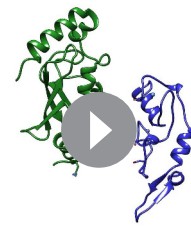

**Video 5.** Salt-bridge competition observed during dUBC13/TRAF6$^{RING}$ association (100 mM NaCl) by conventional MD. The movie shows the order of contact formation during the association of TRAF6$^{RING}$ and dUBC13 (*Figure 8C–D*, *Figure 8—figure supplement 1B*). Initial association occurs through the formation of a non-native R6-E69 salt-bridge due to the presence of the R14-E100 intramolecular salt-bridge. Hydrophobic and S96-P106 contacts are weakly present from ~ 5 to 40 ns. At ~ 22 ns, salt-bridge competition occurs, leading to the formation of the R14-E69 salt-bridge. E100$^{UBC13}$ eventually outcompetes E69$^{TRAF6}$ for R14 which leads to a disruption of the transient complex by 50 ns. Residues R6/R14/S96/E100 in UBC13 (Green) and D57/E69/P106 in TRAF6$^{RING}$ (Blue) are shown in stick representation. A black, dashed-line is drawn between R14 (Cζ) and E100 (Cδ) which indicates the presence/absence of an intramolecular salt-bridge depending on the length of the line.
https://elifesciences.org/articles/49223#video5

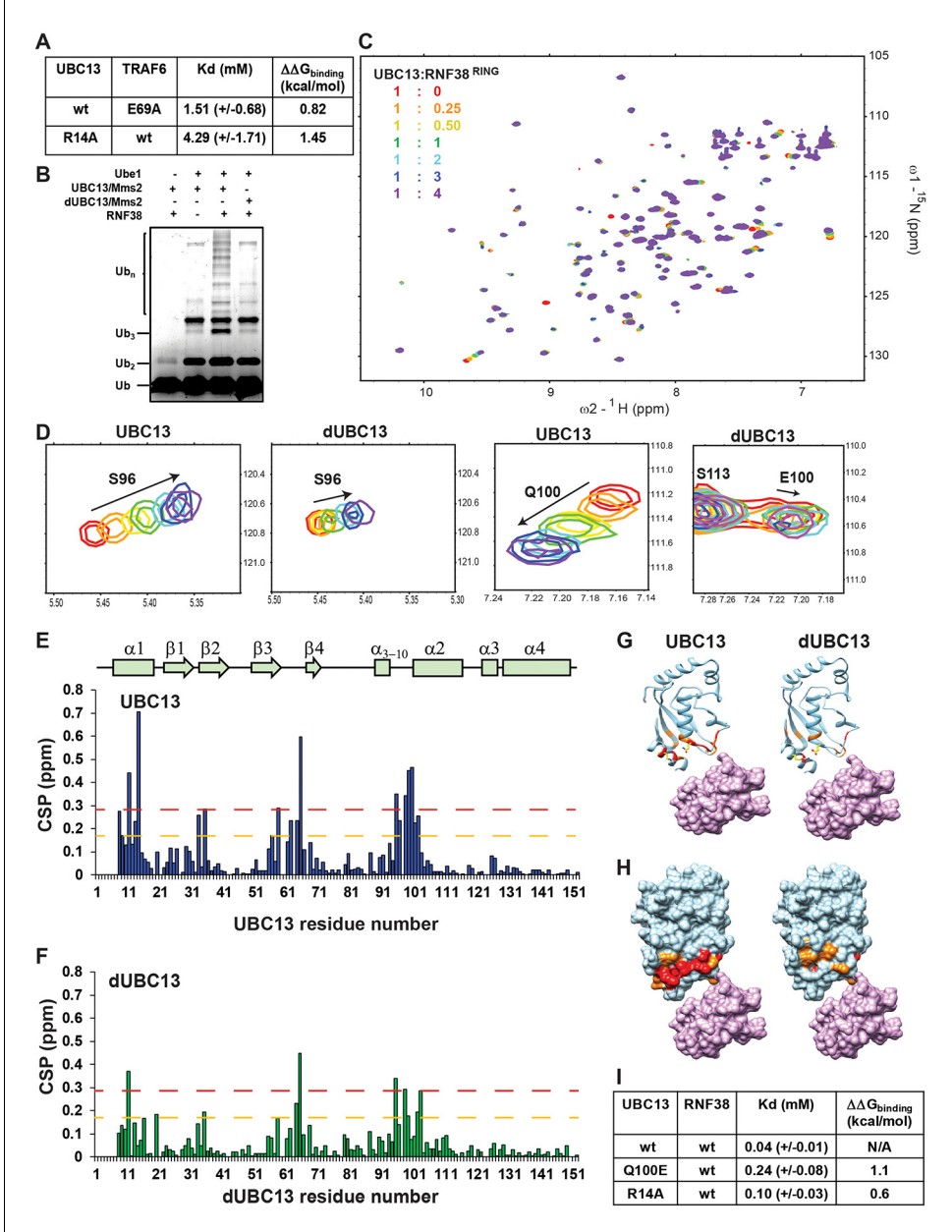

**Figure 9.** Activity and interactions of the UBC13/RNF38$^{RING}$ complex. (**A**) The measured dissociation constants of mutants of UBC13/TRAF6$^{RING}$ complex are given as Mean+/-SD. The difference of free energy of binding was calculated as $\Delta\Delta G_{binding} = RTln(K_d/K_d{}^{wt})$, where T is 298K, and wt is the wild type complex. (**B**) In vitro ubiquitination assay was performed using UBC13/Mms2 (or dUBC13/Mms2) and RNF38. (**C**) Overlay of the $^{15}$N-edited HSQC spectra of free UBC13 (red) with different stoichiometric ratios of RNF38$^{RING}$ as given in the top left-hand side of the spectra. (**D**) Regions of the HSQC spectra are expanded to show the UBC13 and dUBC13 peaks during titration with RNF38$^{RING}$. (**E**) The CSPs for each residue in UBC13 upon binding to RNF38$^{RING}$. The orange and red dashed lines correspond to Mean + SD and Mean + 2*SD, respectively. The secondary structure alignment of UBC13 against its sequence is provided above the plot. (**F**) The CSPs for each residue in dUBC13 upon binding to RNF38$^{RING}$. The dashed lines are replicated from (**E**). Significant CSPs were mapped on the UBC13 and dUBC13 structure using both the (**G**) ribbon and (**H**) surface representation. The UBC13 and dUBC13 are colored in light blue. The residues with CSPs above Mean + SD and Mean + 2*SD are colored in orange and red, respectively. The RNF38$^{RING}$ domain is surface rendered and colored in magenta. (**I**) The measured dissociation constants of UBC13 and its mutants with RNF38$^{RING}$ domain are given as Mean+/-SD. The difference in binding is calculated as in A).

The online version of this article includes the following source data and figure supplement(s) for figure 9:

**Source data 1.** Source data of chemical shift perturbations against UBC13 residue numbers in the UBC13/RNF38$^{RING}$ titration.

**Source data 2.** Source data of chemical shift perturbations against dUBC13 residue numbers in the dUBC13/RING38$^{RING}$ titration.

**Figure supplement 1.** R14-mediated intermolecular salt-bridges observed in crystallographic complexes of UBC13 with RING domains of other E3s.

*Figure 9 continued on next page*

*Figure 9 continued*

**Figure supplement 2.** Analysis of UBC13/RNF38$^{RING}$ structure and charge distribution at and near the interface of RNF38/TRAF6$^{RING}$.

cutoff ($r_{cut}$). The desired value of $r_{cut}$ is determined in an iterative fashion by varying from 0.6 to 1.0 nm (increment of 0.1 nm) so as to generate $10^7$ clash-free configurations. The number of intermolecular contacts (both native and non-native) is calculated for all configurations. The standard deviation in the rotation angle ($\sigma_\chi$) is calculated for configurations at each contact level ($N_c$). The transient complex ensemble corresponds to the value of $N_c$ at which the difference between $\sigma_\chi(N_c)$ and the average of all lower contact levels reaches a maximum.

From the transient complex ensemble, the electrostatic interaction energy ($U_{el}$) was calculated for each configuration as

$$U_{el} = U_{el}(AB) - U_{el}(A) - U_{el}(B) \tag{9}$$

where $U_{el}(AB)$ corresponds to the interaction energy of the complex and $U_{el}(A/B)$ corresponds to that of the two unbound proteins. Electrostatic energies are calculated using the linearized Poisson-Boltzmann equation. $\Delta G_{el}$ is calculated as a simple average over interaction energies of the transient complex ensemble. $k_{a0}$ is determined from 4000 force-free Brownian dynamics trajectories based on a previously developed algorithm (*Zhou, 1993*).

## Videos from MD simulations

All videos were prepared by pre-processing trajectories using a lowpass filter (g_filter tool in Gromacs) followed by the recording of trajectory snapshots in UCSF Chimera using the MD analysis module. In all trajectories, rotational and translational motion of UBC13 was removed entirely to fix its position with respect to TRAF6$^{RING}$.

## Cloning and mutagenesis

A synthetic construct for TRAF6$^{RING}$ (residues 50–124) was synthesized at Thermo Fischer Scientific, USA. PCR amplified ORF was cloned into a pGEX4T1 vector using NEB HF cloning kit (New England Biolabs, Inc USA). E69A-TRAF6 was cloned using PCR based site-directed mutagenesis. Variants of UBC13 were generated using PCR based site-directed mutagenesis in WT-UBC13 in pET24 vector (a gift from Prof. Rachel Klevit). RNF38 plasmid was a gift from Prof. Danny Huang. TRAF6$^{RZ3}$ plasmid was a gift from Prof. Catherine Day. All the plasmids were confirmed by sequencing.

## Protein expression and purification

All proteins were expressed in BL21 DE3 star cells (Invitrogen), grown at 37°C (in LB media for unlabeled proteins and in M9 media for C-13 or N-15 labeled proteins) till OD$_{600}$ reached 0.7 and were induced with 0.5 mM IPTG for 4–5 hr. The harvested cells were lysed in 50 mM Tris, 250 mM NaCl, 5 mM βME, 2% glycerol, 0.01% triton-X, 1 mM PMSF and DNase at pH 7.5 using Emulsiflex high-pressure homogenizer. GST-TRAF6$^{RING}$ and GST-RNF38$^{RING}$ (residues 387–465) were bound to GSTrap HP columns (GE Healthcare) and were eluted with 10 mM reduced glutathione. The proteins were further purified by gel filtration on Superdex 75 pg 16/600 (GE Healthcare) in 25 mM Tris, 100 mM NaCl and 5 mM βME at pH 7.5. The His-tagged UBC13 and its variants were bound to HisTrap HP columns (GE Healthcare) and were eluted with 100 mM-500mM Imidazole gradient. The proteins were further purified by gel filtration on Superdex 75 pg 16/600 (GE Healthcare) in 25 mM Tris, 100 mM NaCl and 5 mM βME at pH 7.5. TRAF6$^{RZ3}$ was induced at OD$_{600}$ ~ 0.7 with 0.2 mM IPTG and 0.1 mM ZnCl$_2$. The lysed cells were resuspended in 50 mM Tris, 350 mM NaCl, 7 mM βME, 1 mM PMSF, 10 mg Lysozyme, 0.01% Triton X, pH 8 and centrifuged. The supernatant was bound to HisTrap beads and was with 150 mM-450mM Imidazole gradient. The eluent was subsequently purified by gel filtration on Superdex 75 pg 16/600 (GE Healthcare) in 25 mM Tris, 100 mM NaCl and 5 mM βME at pH 7.5.

For the purification of Ub and its variants, lysed cells were resuspended in lysis buffer (50 mM Sodium acetate, 5 mM BME, 0.01% Triton X, pH 4.5), lysed by sonication and centrifuged to pellet cell debris. The supernatant was passed through the SP FF column (GE Healthcare) and the protein was eluted by gradient elution with increasing salt concentration (0 mM NaCl- 600 mM NaCl). The

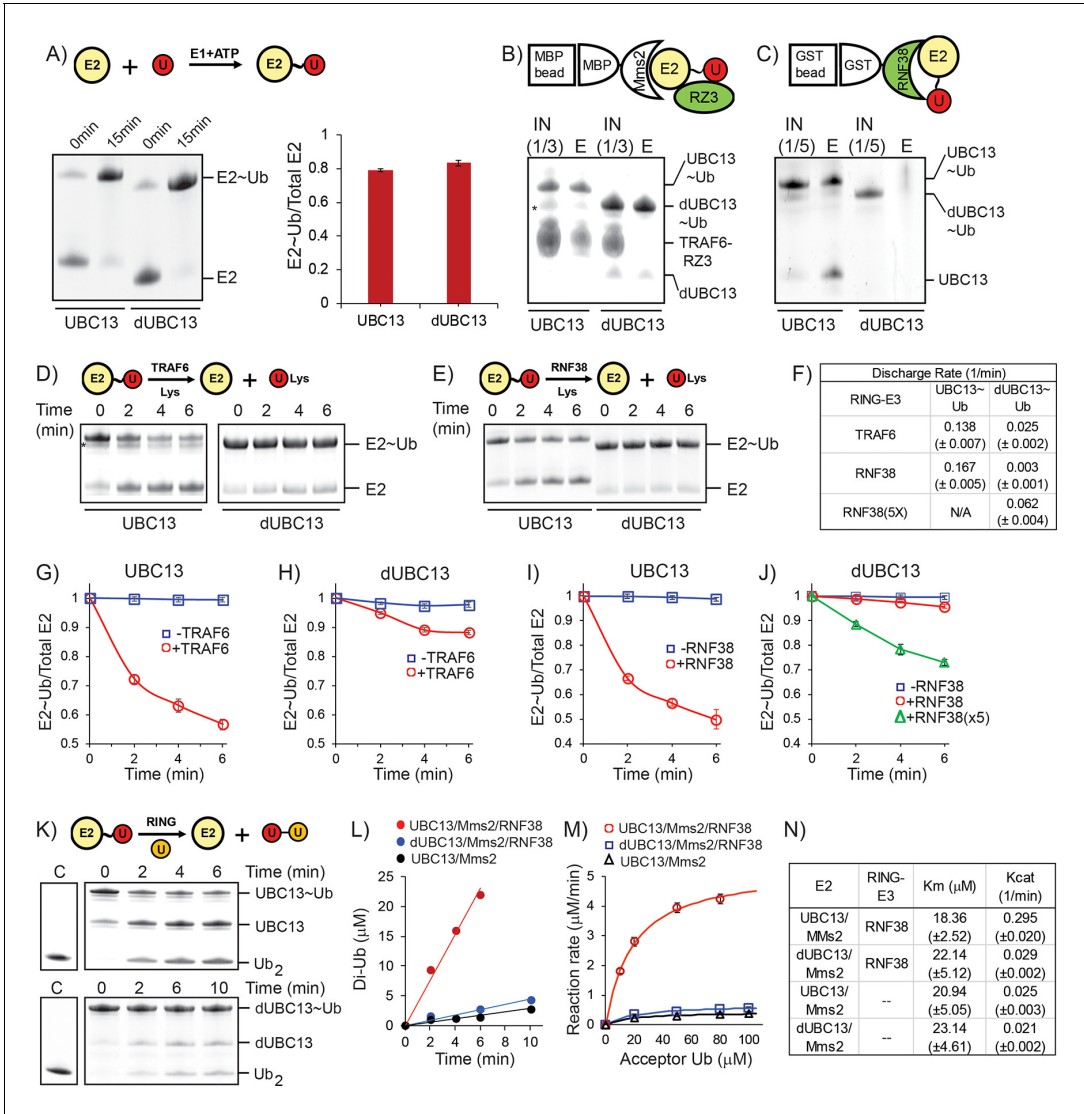

**Figure 10.** Activation of UBC13 ~ Ub conjugates by RING domains. (**A**) A comparison of Ub conjugation to UBC13 and dUBC13. E1, ATP, and UBC13/dUBC13 were incubated in reaction buffer for 15 min, quenched by adding EDTA and separated on SDS page. The amount of E2 ~ Ub conjugates were quantified and plotted in the right section. The values are the mean of three reactions, and the error is the standard deviation of the same. (**B**) Affinity pull-down experiment was performed by incubating MBP beads with MBP-Mms2, UBC13 ~ Ub (or dUBC13 ~ Ub) and TRAF6[RZ3], washed thoroughly and separated on SDS gels. The asterisk denotes impurities. (**C**) Affinity pull-down experiment was performed by incubating GST beads with GST-RNF38[RING] and UBC13 ~ Ub (or dUBC13 ~ Ub), washed thoroughly and separated on SDS gels. (**D**) Single-round discharge of Ub from UBC13 ~ Ub and dUBC13 ~ Ub catalyzed by TRAF6[RING] was monitored. UBC13 was conjugated with Ub, and the reaction was quenched. Then Mms2, TRAF6[RING], and Lysine were added to the reaction mixture, and E2 ~ Ub and free E2 was monitored over time. The proteins bands in (**D**) were quantified and plotted in (**G**) and (**H**). The plotted values are the mean of triplicates, and the error is the standard deviation of the same. (**E**) Same as in D), where the discharge is catalyzed by RNF38[RING] domain. The proteins in (**E**) are quantified and plotted in (**I**) and (**J**). The rate of discharge in each case was calculated as discharge-rate = (Total E2-E2 ~ Ub)/(Total E2.time) for the initial time points and given in (**F**). (**K**) The rate of Ub$_2$ synthesis was monitored over time. UBC13 or dUBC13 was conjugated with Ub, and the reaction was quenched. Then Mms2, RNF38[RING] and Acceptor Ub was added to the reaction mixture, and the synthesis of Ub$_2$ was monitored over time. (**L**) The rate of Ub$_2$ synthesis for UBC13 and dUBC13 are compared at substrate (acceptor-Ub) concentration of 50 µM. (**M**) The reaction rates for UBC13 and dUBC13 are compared at various substrate concentrations. (**N**) The kinetic parameters of Ub$_2$ synthesis for UBC13 and dUBC13 are given in a table. The values are the mean of triplicates, and the error is the standard deviation of the same. The Ub used in all the experiments of this figure is K63A-Ub, except the acceptor Ub used in (**K**)-(**M**) is D77-Ub. The online version of this article includes the following figure supplement(s) for figure 10:

**Figure supplement 1.** Kinetics of Ub conjugation, RING-binding, and Ub-discharge.
**Figure supplement 2.** Kinetics of Ub$_2$ synthesis.
**Figure supplement 3.** Stabilization of the UBC13 ~ Ub conjugate by the TRAF6 homodimer.

proteins were further purified by gel filtration chromatography on Superdex 75 16/600 column (GE Healthcare) in 50 mM Sodium Phosphate, 100 mM NaCl, 5 mM BME, pH 6.5. For the purification of control K63-linked di-Ub, 1 µM UbE1, 40 µM Ubc13, 40 µM GB1-MMS2, 40 µM GST-RNF38, 200 µM D77-Ub and 200 µM K63A-Ub were mixed with reaction buffer (50 mM Sodium Phosphate, 5 mM MgCl$_2$, 3 mM ATP, 0.5 mM DTT, pH 6.5) overnight. The mixture was dialyzed in 50 mM Sodium Acetate, 5 mM BME, pH 4.5, and centrifuged. The supernatant was passed through the SP FF column (GE Healthcare) and the di-Ub was separated by gradient elution with increasing salt concentration (0 mM NaCl- 600 mM NaCl).

## NMR spectroscopy

All NMR titration experiments were recorded at 298K on 600 MHz or 800 MHz Bruker Avance III HD spectrometer with a cryoprobe head. The samples were prepared in 25 mM Tris, 100 mM NaCl, pH 7.5, and 10% D$_2$O. For NMR titration experiments, either ~ 2 mM TRAF6$^{RING}$or ~ 1 mM RNF38$^{RING}$ were titrated into ~ 0.15 mM $^{15}$N-UBC13, $^{15}$N-dUBC13 and other mutants. The titration data was fit in 1:1 protein:ligand model using the equation CSP$_{obs}$ = CSP$_{max}$ {([P]$_t$+[L]$_t$+K$_d$) - [([P]$_t$+[L]$_t$+K$_d$)$^2$- 4 [P]$_t$[L]$_t$]$^{1/2}$}/2[P]$_t$, where [P]$_t$ and [L]$_t$ are total concentrations of protein and ligand at any titration point.

The Arginine sidechain spectra of UBC13 and dUBC13 were recorded at 298K on 800 MHz Bruker Avance III HD spectrometer with a cryoprobe head, processed with NMRpipe (*Delaglio et al., 1995*) and analyzed with Sparky (*Kneller and Kuntz, 1993*). The samples were prepared in 25 mM phosphate, 50 mM NaCl, pH 6.0. D$_2$O was not directly added to the NMR sample to avoid additional signals due to 2H isotopomers. For NMR lock, the 3 mm NMR tube was inserted co-axially in an external tube containing D$_2$O. The arginine sidechain Hε, Nε, Cζ resonances were assigned by broadband NOESY-HSQC, HNCACB (Junji Iwahara and Marius Clore, JBNMR 2006), 2D HD(CD)NE and HD(CDNE)CZ (Frans Mulder JACS 2007) experiments.

## NMR data-driven structural model of UBC13/RNF38$^{RING}$

The solution structure of UBC13/RNF38$^{RING}$ complex was calculated in HADDOCK (*Dominguez et al., 2003*) using the structure of UBC13 (PDB ID: 3HCT) and RNF38$^{RING}$ (PDB ID: 4V3K). Rigid body energy minimization generated one thousand initial complex structures, and the 200 lowest energy structures were selected for torsion angle dynamics and subsequent Cartesian dynamics in an explicit water solvent. Default scaling for energy terms was applied. The interface of proteins was kept semi-flexible during simulated annealing and the water refinement steps. The statistics of the highest scored ensemble are, HADDOCK score: −80.6 ± 0.5, Cluster size: 74, RMSD from the overall lowest-energy structure: 0.8 ± 0.5, Van der Waals energy: −44.4 ± 4.4, Electrostatic energy: −169.9 ± 19.6, Desolvation energy: −8.6 ± 5.1 and Buried Surface Area: 1274.1 ± 75.3.

## In-vitro ubiquitination assay, discharge assay, and Ub$_2$ synthesis assay

For ubiquitination assay using TRAF6$^{RZ3}$, E1 (0.5 µM), UBC13 and MMS2 (20 µM) and Alexa Fluor Maleimide (Invitrogen) labeled UbS20C (20 µM) were incubated with TRAF6$^{RZ3}$ (15 µM) in UB-buffer (50 mM Tris, 5 mM ATP, 5 mM MgCl$_2$,20 mM NaCl, 2 mM DTT, pH 8) at 37˚C for 10 min. The same reaction was repeated with GST-TRAF6$^{RING}$ instead of TRAF6$^{RZ3}$. The reaction mixtures were separated on 12% SDS gel, and the images were acquired in Uvitec (Cambridge). For ubiquitination assay using GST-RNF38$^{RING}$, E1 (0.5 µM), UBC13 and MMS2 (5 µM) and Alexa Fluor Maleimide (Invitrogen) labeled UbS20C (20 µM) were incubated with GST-RNF38$^{RING}$ (3 µM) in 20 mM Tris, 5 mM ATP, 5 mM MgCl$_2$ (pH 7.5) at 37˚C for 30 min. The reaction mixtures were separated in 12%–15% SDS gel, and the images were acquired in iBright FL1000 (Invitrogen). For Ub conjugation assays, E1 (1 µM), UBC13/dUBC13 (20 µM), and K63A-Ub (100 µM) were incubated in UB-buffer at 37˚C for 15 min. For discharge assays, E1 (1 µM), UBC13/dUBC13 (20 µM), and K63A-Ub (100 µM) were incubated in UB-buffer at 37˚C for 60 min. The reaction was quenched with EDTA (30 mM) at room temperature for 5 min. MBP-Mms2 (40 µM), TRAF6$^{RING}$ or RNF38$^{RING}$ (40 µM) and Lysine (15 mM) were added to start Ub-discharge. The reaction was quenched by 4X non-reducing SDS loading buffer at desired time-points. For Ub$_2$ synthesis kinetics, E1 (1 µM), UBC13/dUBC13 (20 µM), and K63A-Ub (100 µM) were incubated in UB-buffer at 37˚C for 60 min. The reaction was quenched with EDTA (30 mM) at room temperature for 5 min. Then MBP-Mms2 (40 µM), RNF38$^{RING}$ (40 µM) and D77-Ub (20–100

µM) were added to start Ub$_2$ synthesis. The reaction was quenched by 4X non-reducing SDS loading buffer at desired time-points. The reaction mixtures of Ub conjugation, Ub discharge, and Ub$_2$ synthesis were separated in 12% SDS gel, stained overnight with SYPRO Ruby (Invitrogen) and the images were acquired in Uvitec (Cambridge).

### In-vitro binding assay

For the pull-down of TRAF6$^{RZ3}$ with UBC13 ~ Ub conjugate, Ub conjugation reaction was performed as described above, and the reaction was quenched with EDTA at room temperature for 5 min. MBP-Mms2 (80 µM) was incubated on amylose bead (NEB) in 50 mM Tris, 20 mM NaCl, pH 8. The UBC13 ~ Ub conjugate (20 µM) was then mixed with TRAF6$^{RZ3}$ (60 µM) and MBP-Mms2 on the amylose beads for 1 hr at 4℃, washed (3x, each time centrifuged at 3500 rpm for 3 min). After the final wash, the pellet was resuspended in 50 mM Tris, 20 mM NaCl, pH 8 and separated on 12% SDS gel. For the pull-down of UBC13 ~ Ub conjugate with GST-RNF38$^{RING}$, the GST-RNF38$^{RING}$ was incubated with glutathione agarose (Thermo Fisher). The UBC13 ~ Ub conjugate was mixed with GST-RNF38$^{RING}$ (60 µM) on glutathione agarose for 1 hr at 4℃, washed (3x), resuspended in 50 mM Tris, 20 mM NaCl, pH 8 and separated on 12% SDS gel. The gels for both pull-down assays were stained overnight with SYPRO Ruby (Invitrogen), and the images were acquired in Uvitec (Cambridge).

## Acknowledgements

The NMR spectra were collected at the NMR Facility and simulations were performed at the Computing facility of the National Centre for Biological Sciences. The authors thank Prof. Rachel Klevit and Prof. Danny Huang, Prof. Catherine Day for reagents, and Dr. Purushotham Reddy for help with data collection. This work was funded by the Tata Institute of Fundamental Research. RD acknowledges the DBT-Ramalingaswamy fellowship (BT/HRD/23/02/2006).

## Additional information

### Funding

| Funder | Grant reference number | Author |
| --- | --- | --- |
| Tata Institute of Fundamental Research | Intramural grant | Ranabir Das |
| Department of Biotechnology, Ministry of Science and Technology | Ramalingaswamy Fellowship | Ranabir Das |

The funders had no role in study design, data collection and interpretation, or the decision to submit the work for publication.

### Author contributions

Priyesh Mohanty, Data curation, Formal analysis, Investigation, Writing-original draft, Writing-review and editing; Rashmi Agrata, Formal Analysis, Investigation, Methodolgy, Writing-review and editing; Batul Ismail Habibullah, Arun G S, Investigation, Methodology; Ranabir Das, Conceptualization, Formal analysis, Supervision, Funding acquisition, Investigation, Writing—original draft, Project administration, Writing—review and editing

### Author ORCIDs

Ranabir Das https://orcid.org/0000-0001-5114-6817

### Decision letter and Author response

Decision letter https://doi.org/10.7554/eLife.49223.SA1
Author response https://doi.org/10.7554/eLife.49223.SA2

## Additional files

### Supplementary files

• Supplementary file 1. Contains six tables to provide salt-bridge occupancies and other analyzed parameters from molecular dynamics simulations.

• Transparent reporting form

### Data availability

All data generated or analysed during this study are included in the manuscript and supporting files.

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
