## [Decision Letter]

**Acceptance summary:**

This study provides a detailed analysis of how hydrolysis of a single amino acid by a pathogenic enzyme can inactivate a ubiquitin E3 ligase. A particular strength of the manuscript is its integration of NMR structural analysis, computational modeling, and well performed enzymatic experiments. This paper stands out for its mechanistic biochemical rigor in explaining how an infectious organism can attenuate the inflammatory response to promote its ability to survive in a human host.

**Decision letter after peer review:**

Thank you for submitting your article "Deamidation disrupts native and transient contacts to weaken the interaction between UBC13 and RING-finger E3 ligases" for consideration by *eLife*. Your article has been reviewed by four peer reviewers, including Lew E Kay as the Reviewing Editor and Reviewer #1, and the evaluation has been overseen by Philip Cole as the Senior Editor.

The reviewers have discussed the reviews with one another and the Reviewing Editor has drafted this decision to help you prepare a revised submission.

All reviewers were impressed with the breadth of the biophysics that was brought to the problem, but several felt that a number of the claims were not sufficiently rigorously supported by the data. In addition, it was felt that the paper was lengthy and very detailed, to the point that it might not receive the interest from the readership that it deserves. In a suitably revised version we ask that the points raised below be adequately addressed, in particular, ensuring that individual claims are clearly supported by associated data.

Major points:

1) In addition to examining the UBC13/TRAF complex they also look at a second interaction involving UBC13/RNF38. The attribute the lack of a 'E69' in RNF38 to indicating that the loss of observed catalysis upon deamidation in this system is due to perturbation of transient complexes. We think more justification of this statement is required. For example, the corresponding Arg that would normally participate in the inter-molecular contact in this case would now be 'free'. Cant this influence the bound complex as well? If this Arg could form an intra-interaction (like R14-E100 in the TRAF complex) then this would 'sequester' the Arg which otherwise (in the Q100 protein) might be available at the interface in the fully bound complex. So we could certainly imagine changes to the final complex as well.

2) The data quite convincingly show a defect caused by replacement of Q100 with an acidic side-chain, and of a mutation in the nearby Arg14. The work is comprehensive and considers both direct effects of mutations and indirect effects of competition for salt bridges. The present study goes to great lengths to provide a physical explanation for the defects caused by mutations. The quality of the work is high, but we feel that the detailed and focused nature of the study will primarily be of interest to specialists who study effects of charge mutations on protein-protein interactions. With this in mind the authors are asked to provide a revised copy that is more cogent so that the general readership can obtain a better appreciation of the work.

3) In general, the methodology is adequate for the purpose of the calculations presented, but the authors should provide more statistical analysis to support the significance of their results. Failing which, some of the conclusions are overstated: "The repulsive interaction destabilized the complex […] and triggered its dissociation. The R14/E100 intramolecular salt-bridge formed soon after dissociation, which would prevent further transient association between R14 and TRAF6 residues." These conclusions are overstated as the authors have not quantified the strength of either salt bridge, the observations of transient contacts are anecdotal (only one window was analyzed in Figure 6), and causality is not established. Similar comments apply to the data shown in Figure 8.

"The intramolecular salt-bridge is dominant and effectively outcompetes the intermolecular salt bridge (Figure 5E)." But Figure 5E only shows that, in one of many simulations, the former replaces the latter at sufficiently long intermolecular distances. In addition, Figure 4D shows that the intermolecular salt bridge is more populated than the intramolecular one in the complex, which argues against the "dominance" of the latter. "Transient complex theory" calculations using the TransComp server are mentioned but not described. They should be described in Materials and methods in enough detail that they can be reproduced by competent researchers.

---

## [Author Response]

All reviewers were impressed with the breadth of the biophysics that was brought to the problem, but several felt that a number of the claims were not sufficiently rigorously supported by the data. In addition, it was felt that the paper was lengthy and very detailed, to the point that it might not receive the interest from the readership that it deserves. In a suitably revised version we ask that the points raised below be adequately addressed, in particular, ensuring that individual claims are clearly supported by associated data.

We thank the experts for their comments/suggestions and their time. We have added additional NMR data, structural data, and MD analysis to ensure that the individual claims are rigorously supported by data. The paper was condensed by moving technical details to the Figure legends or the Materials and methods section. The text has been changed at multiple places to improve coherence. The response to each comment is given below.

Major points:1) In addition to examining the UBC13/TRAF complex they also look at a second interaction involving UBC13/RNF38. The attribute the lack of a 'E69' in RNF38 to indicating that the loss of observed catalysis upon deamidation in this system is due to perturbation of transient complexes. We think more justification of this statement is required. For example, the corresponding Arg that would normally participate in the inter-molecular contact in this case would now be 'free'. Cant this influence the bound complex as well? If this Arg could form an intra-interaction (like R14-E100 in the TRAF complex) then this would 'sequester' the Arg which otherwise (in the Q100 protein) might be available at the interface in the fully bound complex. So we could certainly imagine changes to the final complex as well.

We appreciate the comment raised by the reviewer. To address this point, we have purified the ^13^C,^15^N-labeled labeled RNF38^RING^ and assigned the chemical shifts of backbone N, HN, Cα, and C’ atoms. Then, we carried out NMR titrations to map the binding interface of RNF38^RING^ in the UBC13/RNF38^RING^ complex. Combining this new data with the UBC13 interface mapped earlier, we calculated NMR-data driven docked structure of the UBC13/RNF38^RING^ complex by the software HADDOCK.

In this complex, we find that R14 makes a van der Waals interaction with M417 of RNF38^RING^. However, this contact is no more a ‘hot-spot’ contact. To mimic the sequestration of R14, we made R14A substitution in the structure, which disrupted the R14/M147 native contact. The binding energy calculations of native complex suggested that R14A substitution, i.e., disruption of the R14 native contact, changes the binding energy of the complex by only 0.2 kcal/mol. In contrast, the dissociation constants measured by NMR indicate a change of binding energy by 1.1 kcal/mol upon deamidation and 0.6 kcal/mol upon R14A substitution (Table 9I). Hence, apart from the R14 native contact, the R14 transient contacts also have a 0.4kcal/mol energy contribution to the binding. Deamidation induced salt-bridge sequestration will hamper both these interactions.

The new experiments and analysis are now mentioned in Figure 9—figure supplement 2 and in the text in subsection “The effect of deamidation on transient interactions persist in the UBC13/RNF38RING complex”. Details of the structure calculation of the UBC13/RNF38^RING^ complex is added to the Materials and methods.

2) The data quite convincingly show a defect caused by replacement of Q100 with an acidic side-chain, and of a mutation in the nearby Arg14. The work is comprehensive and considers both direct effects of mutations and indirect effects of competition for salt bridges. The present study goes to great lengths to provide a physical explanation for the defects caused by mutations. The quality of the work is high, but we feel that the detailed and focused nature of the study will primarily be of interest to specialists who study effects of charge mutations on protein-protein interactions. With this in mind the authors are asked to provide a revised copy that is more cogent so that the general readership can obtain a better appreciation of the work.

Indeed, the mechanism boils down to atomistic details, but the effect is relevant in the context of how one PTM signaling can modulate another PTM signaling, and also in the context of host-Shigella interactions. We have now condensed the text by moving technical details to Figure captions and Materials and methods. The text is changed in several places to explain the implications of analysis and appeal to the general readership.

3) In general, the methodology is adequate for the purpose of the calculations presented, but the authors should provide more statistical analysis to support the significance of their results. Failing which, some of the conclusions are overstated: "The repulsive interaction destabilized the complex […] and triggered its dissociation. The R14/E100 intramolecular salt-bridge formed soon after dissociation, which would prevent further transient association between R14 and TRAF6 residues." These conclusions are overstated as the authors have not quantified the strength of either salt bridge, the observations of transient contacts are anecdotal (only one window was analyzed in Figure 6), and causality is not established.

To provide statistics and strength of salt-bridges, we have now provided salt-bridge occupancies for the conventional MD (Table S1), SMD (Table S4), US window (Table S5) and unbiased MD (Table S6) trajectories in Supplementary File 1. Table S1 shows that the strength of intramolecular salt-bridge drops from 99% to 40% upon deamidation, and the new intermolecular salt-bridge strength ~30%, suggesting that the two salt-bridges can compete. The salt-bridge occupancies during dissociation in SMD also confirmed that the two salt-bridges compete (Tables S4). The salt-bridge occupancies in the native window of US simulations also showed that the intramolecular salt-bridge occupancy drops from 100% to 37% upon deamidation, whereas the intermolecular salt-bridge occupancy in 55% (Table S5). Where native complexes were observed in unbiased association simulations, the intermolecular salt-bridge occupancy reduced from 34% to 1%, and the intermolecular salt-bridge occupancy was 86%. Altogether, the salt-bridge occupancies in multiple MD simulation techniques strongly suggest that the two salt-bridges compete.

To address the comment regarding Figure 6, the E100/TRAF6 Coulombic Interaction Energy were analyzed at multiple US windows from 2.7- 3.1 nm (Figure 6—figure supplement 1A). The native window is at 2.7 nm, and the Coulombic Interaction Energy values are zero beyond 3.1 nm. Hence, US windows beyond 3.1 nm were not considered. The analysis showed that repulsive interactions (+ve Coulombic Interaction Energy values) are consistently observed between E100 and TRAF6^RING^.

Taking multiple US windows between 2.7 nm and 3.1 nm (apart from the native 2.7 nm window shown in Figure 3 and the 3.0 nm window shown in Figure 6), we also show that the deamidated complex is unstable with unfavorable Coulombic interaction energies (Figure 6—figure supplement 1B-E). Hence, the analysis of multiple US windows confirms the presence of repulsive transient interactions between E100 and TRAF6^RING^ and the instability of the deamidated complex. The new MD analysis is added as Figure 6—figure supplement 1 and discussed in subsection “Repulsive interactions between E100 and TRAF6RING destabilize the transient complex”.

Similar comments apply to the data shown in Figure 8.

For the high-affinity complexes, the probability of capturing native complexes in unbiased association simulations are higher. However, for low-affinity E2/RING complexes, we typically observe few trajectories which capture the native complex, making the statistical analysis difficult. However, for the trajectories corresponding to the native complexes shown in Figure 8, we now have provided the salt-bridge occupancies in Table S6. Here, the intermolecular salt-bridge occupancy reduced from 34% to 1%, and the intermolecular salt-bridge occupancy was 86%, indicating the salt-bridge competition. We have changed the text and figure legends to reflect that the unbiased association events are observations and do not imply causality.

"The intramolecular salt-bridge is dominant and effectively outcompetes the intermolecular salt bridge (Figure 5E)." But Figure 5E only shows that, in one of many simulations, the former replaces the latter at sufficiently long intermolecular distances.

In Figure 5E, the intramolecular replaces the intermolecular at 3 ns, whereas the complex starts to dissociate at 5 ns as shown by the force-profile in Figure 4A. Hence, the intermolecular distance is small.

We agree that the intermolecular salt-bridge may not dominate but effectively competes with the intramolecular salt-bridge. The sentence in the discussion is deleted. We have changed the sentence on to reflect this:

“However, in this trajectory of the dUBC13 complex trajectory, the intermolecular salt-bridge disrupted early, and the intramolecular salt-bridge formed simultaneously (Figure 5D-5F). Collectively, the SMD suggested that R14/E69 formed a critical interfacial salt-bridge, and as the molecules start to dissociate, the R14/E69 and R14/E100 salt-bridges compete. The competition may contribute to the reduced stability of dUBC13 complex.”

In addition, Figure 4D shows that the intermolecular salt bridge is more populated than the intramolecular one in the complex, which argues against the "dominance" of the latter.

We want to show that as the TRAF6 starts dissociating, the intermolecular salt-bridge competes. This was unclear in the previous version. Hence, we have reanalyzed Figure 4D for the duration of 5 ns – 15 ns. At 5 ns, TRAF6 begins to dissociate, and the intermolecular salt-bridge persists for a maximum of 15 ns. The new analysis shows that the two salt-bridges compete during this period (Figure 4D). The occupancies of the salt-bridges during this period is now provided as a table (Table S4). A sentence is also added to explain this:

“As the TRAF6^RING^ starts to dissociate, the R14/E100 salt-bridge competes with the R14/E69 salt-bridge and reduces its stability (Figure 4D, Figure 4—figure supplement 1, 2).”

"Transient complex theory" calculations using the TransComp server are mentioned but not described. They should be described in Materials and methods in enough detail that they can be reproduced by competent researchers.

The transient complex theory calculations are now described in Materials and methods in detail.